# Diffusion Model is an Effective Planner and Data Synthesizer for Multi-Task Reinforcement Learning

**Haoran He** [1,2*]   **Chenjia Bai** [2†]   **Kang Xu** [2,3]   **Zhuoran Yang** [4]   **Weinan Zhang** [1]

**Dong Wang** [2]   **Bin Zhao** [2,5]   **Xuelong Li** [2,5†]

[1]Shanghai Jiao Tong University   [2]Shanghai Artificial Intelligence Laboratory
[3]Fudan University   [4]Yale University   [5]Northwestern Polytechnical University

## Abstract

Diffusion models have demonstrated highly-expressive generative capabilities in vision and NLP. Recent studies in reinforcement learning (RL) have shown that diffusion models are also powerful in modeling complex policies or trajectories in offline datasets. However, these works have been limited to single-task settings where a generalist agent capable of addressing multi-task predicaments is absent. In this paper, we aim to investigate the effectiveness of a single diffusion model in modeling large-scale multi-task offline data, which can be challenging due to diverse and multimodal data distribution. Specifically, we propose Multi-Task Diffusion Model (MTDIFF), a diffusion-based method that incorporates Transformer backbones and prompt learning for generative planning and data synthesis in multi-task offline settings. MTDIFF leverages vast amounts of knowledge available in multi-task data and performs implicit knowledge sharing among tasks. For generative planning, we find MTDIFF outperforms state-of-the-art algorithms across 50 tasks on Meta-World and 8 maps on Maze2D. For data synthesis, MTDIFF generates high-quality data for testing tasks given a single demonstration as a prompt, which enhances the low-quality datasets for even unseen tasks.

## 1   Introduction

The high-capacity generative models trained on large, diverse datasets have demonstrated remarkable success across vision and language tasks. An impressive and even preternatural ability of these models, e.g. large language models (LLMs), is that the learned model can generalize among different tasks by simply conditioning the model on instructions or prompts [49, 46, 7, 12, 60, 42, 66]. The success of LLMs and vision models inspires us to utilize the recent generative model to learn from large-scale offline datasets that include multiple tasks for generalized decision-making in reinforcement learning (RL). Thus far, recent attempts in offline decision-making take advantage of the generative capability of diffusion models [54, 20] to improve long-term planning [21, 2] or enhance the expressiveness of policies [63, 11, 8]. However, these works are limited to small-scale datasets and single-task settings where broad generalization and general-purpose policies are not expected. In multi-task offline RL which considers learning a single model to solve multi-task problems, the dataset often contains noisy, multimodal, and long-horizon trajectories collected by various policies across tasks and with various qualities, which makes it more challenging to learn policies with broad generalization and transferable capabilities. Gato [47] and other generalized agents [29, 65] take transformer-based architecture

---

*The work was conducted during the internship of Haoran He at Shanghai Artificial Intelligence Laboratory.
†Corresponding authors: Chenjia Bai (baichenjia@pjlab.org.cn), Xuelong Li

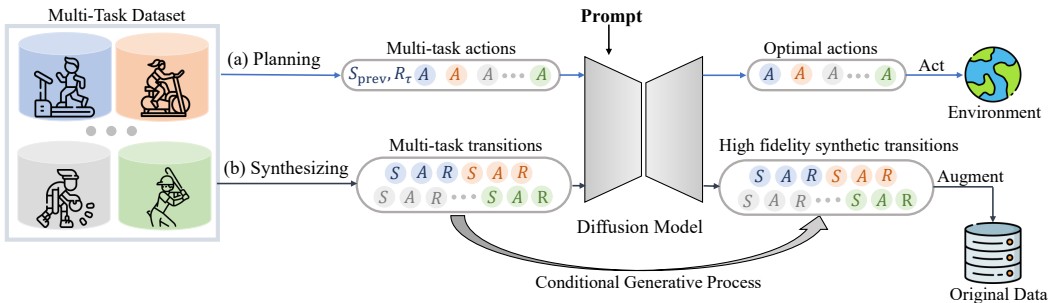

Figure 1: Overall architecture of MTDIFF. Different colors represent different tasks. $S$, $A$ and $R$ denote the state sequence, action sequence, and reward sequence from multi-task data, respectively. $S_{\mathrm{prev}}$ and $R_\tau$ represent historical states and normalized return.

[62] via sequence modeling to solve multi-task problems, while they are highly dependent on the optimality of the datasets and are expensive to train due to the huge number of parameters.

To address the above challenges, we propose a novel diffusion model to further explore its generalizability in a multi-task setting. We formulate the learning process from multi-task data as a denoising problem, which benefits the modeling of multimodal data. Meanwhile, we develop a relatively lightweight architecture by using a GPT backbone [45] to model sequential trajectories, which has less computation burden and improved sequential modeling capability than previous U-Net-based [48] diffusion models [21, 33]. To disambiguate tasks during training and inference, instead of providing e.g. one-hot task identifiers, we leverage demonstrations as *prompt* conditioning, which exploits the few-shot abilities of agents [50, 64, 69].

We name our method the Multi-Task Diffusion Model (**MTDIFF**). As shown in Figure 1, we investigate two variants of MTDIFF for planning and data synthesis to further exploit the utility of diffusion models, denoted as **MTDIFF-P** and **MTDIFF-S**, respectively. (a) For planning, MTDIFF-P learns a prompt embedding to extract the task-relevant representation, and then concatenates the embedding with the trajectory's normalized return and historical states as the *conditions* of the model. During training, MTDIFF-P learns to predict the corresponding future action sequence given the conditions, and we call this process generative planning [78]. During inference, given few-shot prompts and the desired return, MTDIFF-P tries to denoise out the optimal action sequence starting from the current state. Surprisingly, MTDIFF-P can adapt to unseen tasks given well-constructed prompts that contain task information. (b) By slightly changing the inputs and training strategy, we can unlock the abilities of our diffusion model for data synthesis. Our insight is that the diffusion model, which compresses informative multi-task knowledge well, is more effective and generalist than previous methods that only utilize single-task data for augmentation [38, 52, 28]. Specifically, MTDIFF-S learns to estimate the joint conditional distribution of the full transitions that contain states, actions, and rewards based on the task-oriented prompt. Different from MTDIFF-P, MTDIFF-S learns to synthesize data from the underlying dynamic environments for each task. Thus MTDIFF-S only needs prompt conditioning to identify tasks. We empirically find that MTDIFF-S synthesizes high-fidelity data for multiple tasks, including both seen and unseen ones, which can be further utilized for data augmentation to expand the offline dataset and enhance policy performance.

To summarize, MTDIFF is a diffusion-based method that leverages the multimodal generative ability of diffusion models, the sequential modeling capability of GPT architecture, and the few-shot generalizability of prompt learning for multi-task RL. To the best of our knowledge, we are the first to achieve both effective planning and data synthesis for multi-task RL via diffusion models. Our contributions include: (i) we propose MTDIFF, a novel GPT-based diffusion model that illustrates the supreme effectiveness in multi-task trajectory modeling for both planning and data synthesis; (ii) we incorporate prompt learning into the diffusion framework to learn to generalize across different tasks and even adapt to unseen tasks; (iii) our experiments on Meta-World and Maze2D benchmarks demonstrate that MTDIFF is an effective planner to solve the multi-task problem, and also a powerful data synthesizer to augment offline datasets in the seen or unseen tasks.

## 2 Preliminaries

### 2.1 Reinforcement Learning

**MDP and Multi-task MDP.** A Markov Decision Process (MDP) is defined by a tuple $(\mathcal{S}, \mathcal{A}, \mathcal{P}, \mathcal{R}, \mu, \gamma)$, where $\mathcal{S}$ is the state space, $\mathcal{A}$ is the action space, $\mathcal{P} : \mathcal{S} \times \mathcal{A} \to \mathcal{S}$ is the transition function, $\mathcal{R} : \mathcal{S} \times \mathcal{A} \times \mathcal{S} \to \mathbb{R}$ is the reward function for any transition, $\gamma \in (0, 1]$ is a discount factor, and $\mu$ is the initial state distribution. At each timestep $t$, the agent chooses an action $a_t$ by following the policy $\pi : \mathcal{S} \to \Delta_{\mathcal{A}}$. Then the agent obtains the next state $s_{t+1}$ and receives a scalar reward $r_t$. In single-task RL, the goal is to learn a policy $\pi^* = \arg\max_\pi \mathbb{E}_{a_t \sim \pi} \left[ \sum_{t=0}^{\infty} \gamma^t r_t \right]$ by maximizing the expected cumulative reward of the corresponding task.

In a multi-task setting, different tasks can have different reward functions, state spaces and transition functions. We consider all tasks to share the same action space with the same embodied agent. Given a specific task $\mathcal{T} \sim p(\mathcal{T})$, a task-specified MDP can be defined as $(\mathcal{S}^{\mathcal{T}}, \mathcal{A}, \mathcal{P}^{\mathcal{T}}, \mathcal{R}^{\mathcal{T}}, \mu^{\mathcal{T}}, \gamma)$. Instead of solving a single MDP, the goal of multi-task RL is to find an optimal policy that maximizes expected return over all the tasks: $\pi^* = \arg\max_\pi \mathbb{E}_{\mathcal{T} \sim p(\mathcal{T})} \mathbb{E}_{a_t \sim \pi^{\mathcal{T}}} \left[ \sum_{t=0}^{\infty} \gamma^t r_t^{\mathcal{T}} \right]$.

**Multi-Task Offline Decision-Making.** In offline decision-making, the policy is learned from a static dataset of transitions $\{(s_j, a_j, s'_j, r_j)\}_{j=1}^N$ collected by an unknown behavior policy $\pi_\beta$ [30]. In the multi-task offline RL setting, the dataset $\mathcal{D}$ is partitioned into per-task subsets as $\mathcal{D} = \cup_{i=1}^N \mathcal{D}_i$, where $\mathcal{D}_i$ consists of experiences from task $\mathcal{T}_i$. The key issue of RL in the offline setting is the distribution shift problem caused by temporal-difference (TD) learning. In our work, we extend the idea of Decision Diffuser [2] by considering multi-task policy learning as a conditional generative process without fitting a value function. The insight is to take advantage of the powerful distribution modeling ability of diffusion models for multi-task data, avoiding facing the risk of distribution shift.

Offline RL learns policies from a static dataset, which makes the quality and diversity of the dataset significant [15]. One can perform data perturbation [28] to up-sample the offline dataset. Alternatively, we synthesize new transitions $(s, a, s', r)$ by capturing the underlying MDP of a given task via diffusion models, which expands the original dataset and leads to significant policy improvement.

### 2.2 Diffusion Models

We employ diffusion models to learn from multi-task data $\mathcal{D} = \cup_{i=1}^N \mathcal{D}_i$ in this paper. With $\tau$ the sampled trajectory from $\mathcal{D}$, we denote $\boldsymbol{x}_k(\tau)$ as the $k$-step denoised output of the diffusion model, and $\boldsymbol{y}(\tau)$ is the condition which represents the task attributes and the trajectory's optimality (e.g., returns). A forward diffusion chain gradually adds noise to the data $\boldsymbol{x}_0(\tau) \sim q(\boldsymbol{x}(\tau))$ in $K$ steps with a pre-defined variance schedule $\beta_k$, which can be expressed as

$$q(\boldsymbol{x}_k(\tau)|\boldsymbol{x}_{k-1}(\tau)) := \mathcal{N}(\boldsymbol{x}_k(\tau); \sqrt{1 - \beta_k}\boldsymbol{x}_{k-1}(\tau), \beta_k \boldsymbol{I}). \tag{1}$$

In this paper, we adopt Variance Preserving (VP) beta schedule [67] and define $\beta_k = 1 - \exp\left(-\beta_{min}(\frac{1}{K}) - 0.5(\beta_{\max} - \beta_{\min})\frac{2k-1}{K^2}\right)$, where $\beta_{\max} = 10$ and $\beta_{\min} = 0.1$ are constants. A trainable reverse diffusion chain, constructed as $p_\theta(\boldsymbol{x}_{k-1}(\tau)|\boldsymbol{x}_k(\tau), \boldsymbol{y}(\tau)) := \mathcal{N}(\boldsymbol{x}_{k-1}(\tau)|\mu_\theta(\boldsymbol{x}_k(\tau), \boldsymbol{y}(\tau), k), \Sigma_k)$, can be optimized by a simplified surrogate loss [20]:

$$\mathcal{L}_{\text{denoise}} := \mathbb{E}_{k \sim \mathcal{U}(1,K), \boldsymbol{x}_0(\tau) \sim q, \epsilon \sim \mathcal{N}(\boldsymbol{0}, \boldsymbol{I})}[\left\| \epsilon - \epsilon_\theta(\boldsymbol{x}_k(\tau), \boldsymbol{y}(\tau), k) \right\|^2], \tag{2}$$

where $\epsilon_\theta$ parameterized by a deep neural network is trained to predict the noise $\epsilon \sim \mathcal{N}(\boldsymbol{0}, \boldsymbol{I})$ added to the dataset sample $\boldsymbol{x}_0(\tau)$ to produce $\boldsymbol{x}_k(\tau)$. By setting $\alpha_k := 1 - \beta_k$ and $\bar{\alpha}_k := \prod_{s=1}^k \alpha_s$, we obtain

$$\boldsymbol{x}_{k-1}(\tau) \leftarrow \frac{1}{\sqrt{\alpha_k}}\left(\boldsymbol{x}_k(\tau) - \frac{\beta_k}{\sqrt{1 - \bar{\alpha}_k}}\epsilon_\theta(\boldsymbol{x}_k(\tau), \boldsymbol{y}(\tau), k)\right) + \sqrt{\beta_k}\sigma, \ \sigma \sim \mathcal{N}(\boldsymbol{0}, \boldsymbol{I}), \text{ for } k = \{K, ..., 1\}.$$

Classifier-free guidance [19] aims to learn the conditional distribution $q(\boldsymbol{x}(\tau)|\boldsymbol{y}(\tau))$ without separately training a classifier. In the training stage, this method needs to learn both a conditional $\epsilon_\theta(\boldsymbol{x}_k(\tau), \boldsymbol{y}(\tau), k)$ and an unconditional $\epsilon_\theta(\boldsymbol{x}_k(\tau), \varnothing, k)$ model, where $\boldsymbol{y}(\tau)$ is dropped out. Then the perturbed noise $\epsilon_\theta(\boldsymbol{x}_k(\tau), \varnothing, k) + \alpha(\epsilon_\theta(\boldsymbol{x}_k(\tau), \boldsymbol{y}(\tau), k) - \epsilon_\theta(\boldsymbol{x}_k(\tau), \varnothing, k))$ is used to generate samples latter, where $\alpha$ can be recognized as the guidance scale.

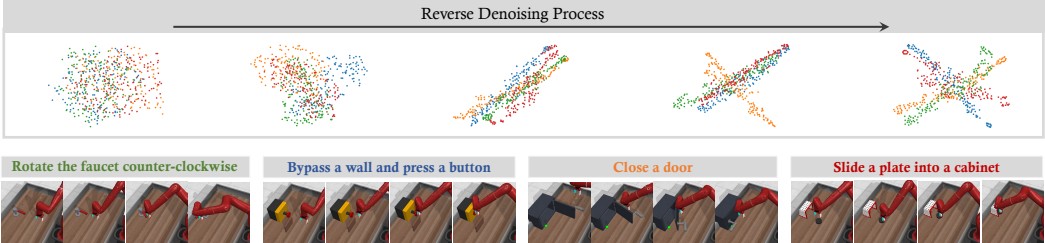

Figure 2: An example of the denoising process of MTDIFF. We choose 4 tasks for visualization and $\boldsymbol{x}_K(\tau)$ is sampled from the Gaussian noise for each task. Since different tasks require different manipulation skills, the corresponding action sequences are dispersed in the embedding space. Our model learns such properties and generates task-specific sequences based on task-relevant prompts.

## 3 Methodology

### 3.1 Diffusion Formulation

To capture the multimodal distribution of the trajectories sampled from multiple MDPs, we formulate the multi-task trajectory modeling as a conditional generative problem via diffusion models:

$$\max_\theta \mathbb{E}_{\tau \sim \cup_i \mathcal{D}_i} \big[ \log p_\theta(\boldsymbol{x}_0(\tau) \mid \boldsymbol{y}(\tau) \big], \tag{3}$$

where $\boldsymbol{x}_0(\tau)$ is the generated desired sequence and $\boldsymbol{y}(\tau)$ is the condition. $\boldsymbol{x}_0(\tau)$ will then be used for generative planning or data synthesis through conditional reverse denoising process $p_\theta$ for specific tasks. Maximizing Eq. (3) can be approximated by maximizing a variational lower bound [20].

In terms of different inputs and outputs in generative planning and data synthesis, $\boldsymbol{x}(\tau)$ can be represented in different formats. We consider two choices to formulate $\boldsymbol{x}(\tau)$ in MTDIFF-P and MTDIFF-S, respectively. (i) For **MTDIFF-P**, $\boldsymbol{x}(\tau)$ represents the action sequence for planning. We model the action sequence defined as:

$$\boldsymbol{x}_k^p(\tau) := (a_t, a_{t+1}, ..., a_{t+H-1})_k, \tag{4}$$

with the context condition as

$$\boldsymbol{y}^p(\tau) := \big[\boldsymbol{y}'(\tau), R(\tau)\big], \qquad \boldsymbol{y}'(\tau) := (Z, s_{t-L+1}, ..., s_t), \tag{5}$$

where $t$, $H$, $R(\tau)$ and $L$ denote the time visited in trajectory $\tau$, the length of the input sequence $\boldsymbol{x}$, the normalized cumulative return under $\tau$ and the length of the observed state history, respectively. $Z$ is the task-relevant information as *prompt*. We use $\boldsymbol{y}'(\tau)$ as an ordinary condition that is injected into the model during both training and testing, while considering $R(\tau)$ as the classifier-free guidance to obtain the optimal action sequence for a given task. (ii) For data synthesis in **MTDIFF-S**, the inputs and outputs become the transition sequence that contains states, actions, and rewards, and then the outputs are utilized for data augmentation. We define the transition sequence as:

$$\boldsymbol{x}_k^s(\tau) := \begin{bmatrix} s_t & s_{t+1} & \cdots & s_{t+H-1} \\ a_t & a_{t+1} & \cdots & a_{t+H-1} \\ r_t & r_{t+1} & \cdots & r_{t+H-1} \end{bmatrix}, \tag{6}$$

with the condition:

$$\boldsymbol{y}^s(\tau) := [Z], \tag{7}$$

where $\boldsymbol{y}^s(\tau)$ takes the same conditional approach as $y'(\tau)$. Figure 2 illustrates the reverse denoising process of MTDIFF-P learned on multi-task datasets collected in Meta-World [73]. The result demonstrates that our diffusion model successfully distinguishes different tasks and finally generates the desired $\boldsymbol{x}_0(\tau)$. We illustrate the data distribution of $\boldsymbol{x}_0(\tau)$ in a 2D space with dimensional reduction via T-SNE [61], as well as the rendered states after executing the action sequence. The result shows that, with different task-specific prompts as conditions, the generated planning sequence for a specific task will be separate from sequences of other tasks, which verifies that MTDIFF can learn the distribution of multimodal trajectories based on $\boldsymbol{y}(\tau)$.

## 3.2 Prompt, Training and Sampling

In multi-task RL and LLM-driven decision-making, existing works use one-hot task identifiers [56, 74] or language descriptions [1, 6] as conditions in multi-task training. Nevertheless, we argue that the one-hot encoding for each task [22, 14] suffices for learning a repertoire of training tasks while cannot generalize to novel tasks since it does not leverage semantic similarity between tasks. In addition, the language descriptions [51, 1, 6, 41, 40] of tasks require large amounts of human labor to annotate and encounter challenges related to ambiguity [72]. In MTDIFF, we use expert demonstrations consisting of a few trajectory segments to construct more expressive prompts in multi-task settings. The incorporation of prompt learning improves the model's ability for generalization and extracting task-relevant information to facilitate both generative planning and data synthesis. We remark that a similar method has also been used in PromptDT [69]. Nonetheless, how such a prompt contributes within a diffusion-based framework remains to be investigated.

Specifically, we formulate the task-specific label $Z$ as trajectory prompts that contain states and actions:

$$Z := \begin{bmatrix} s_i^* & s_{i+1}^* & \cdots & s_{i+J-1}^* \\ a_i^* & a_{i+1}^* & \cdots & a_{i+J-1}^* \end{bmatrix}, \tag{8}$$

where each element with star-script is associated with a trajectory prompt, and $J$ is the number of environment steps for identifying tasks. With the prompts as conditions, MTDIFF can specify the task by implicitly capturing the transition model and the reward function stored in the prompts for better generalization to unseen tasks without additional parameter-tuning.

In terms of decision-making in MTDIFF-P, we aim to devise the optimal behaviors that maximize return. Our approach is to utilize the diffusion model for action planning via classifier-free guidance [19]. Formally, an optimal action sequence $\boldsymbol{x}_0^p(\tau)$ is sampled by starting with Gaussian noise $\boldsymbol{x}_K(\tau)$ and refining $\boldsymbol{x}_k^p(\tau)$ into $\boldsymbol{x}_{k-1}^p(\tau)$ at each intermediate timestep with the perturbed noise:

$$\epsilon_\theta\big(\boldsymbol{x}_k^p(\tau), \boldsymbol{y}'(\tau), \varnothing, k\big) + \alpha\big(\epsilon_\theta(\boldsymbol{x}_k^p(\tau), \boldsymbol{y}'(\tau), R(\tau), k) - \epsilon_\theta(\boldsymbol{x}_k^p(\tau), \boldsymbol{y}'(\tau), \varnothing, k)\big), \tag{9}$$

where $\boldsymbol{y}'(\tau)$ is defined in Eq. (5). $R(\tau)$ is the normalized return of $\tau$, and $\alpha$ is a hyper-parameter that seeks to augment and extract the best portions of trajectories in the dataset with high return. During training, we follow DDPM [20] as well as classifier-free guidance [19] to train the reverse diffusion process $p_\theta$, parameterized through the noise model $\epsilon_\theta$, with the following loss:

$$\mathcal{L}^p(\theta) := \mathbb{E}_{k \sim \mathcal{U}(1,K), \boldsymbol{x}_0(\tau) \sim q, \epsilon \sim \mathcal{N}(\boldsymbol{0}, \boldsymbol{I}), \beta \sim \text{Bern(p)}} \big[\big\| \epsilon - \epsilon_\theta\big(\boldsymbol{x}_k^p(\tau), \boldsymbol{y}'(\tau), (1-\beta)R(\tau) + \beta\varnothing, k\big)\big\|^2\big]. \tag{10}$$

Note that with probability $p$ sampled from a Bernoulli distribution, we ignore the conditioning return $R(\tau)$. During inference, we adopt the *low-temperature sampling* technique [2] to produce high-likelihood sequences. We sample $\boldsymbol{x}_{k-1}^p(\tau) \sim \mathcal{N}(\mu_\theta(\boldsymbol{x}_{k-1}^p, \boldsymbol{y}'(\tau), R_{\max}(\tau), k-1), \beta\Sigma_{k-1})$, where the variance is reduced by $\beta \in [0, 1)$ for generating action sequences with higher optimality.

For MTDIFF-S, since the model aims to synthesize diverse trajectories for data augmentation, which does not need to take any guidance like $R(\tau)$, we have the following loss:

$$\mathcal{L}^s(\theta) := \mathbb{E}_{k \sim \mathcal{U}(1,K), \boldsymbol{x}_0(\tau) \sim q, \epsilon \sim \mathcal{N}(\boldsymbol{0}, \boldsymbol{I})} \big[\big\| \epsilon - \epsilon_\theta(\boldsymbol{x}_k^s(\tau), \boldsymbol{y}^s(\tau), k)\big\|^2\big]. \tag{11}$$

We sample $\boldsymbol{x}_{k-1}^s(\tau) \sim \mathcal{N}(\mu_\theta(\boldsymbol{x}_{k-1}^s, \boldsymbol{y}^s(\tau), k-1), \Sigma_{k-1})$. The evaluation process is given in Fig. 3.

## 3.3 Architecture Design

Notably, the emergence of Transformer [62] and its applications on generative modeling [44, 5, 47, 6] provides a promising solution to capture interactions between modalities of different tasks. Naturally, instead of U-Net [48] which is commonly used in previous single-task diffusion RL works [21, 2, 33], we parameterize $\epsilon_\theta$ with a novel transformer architecture. We adopt GPT2 [45] architecture for implementation, which excels in sequential modeling and offers a favorable balance between performance and computational efficiency. Our key insight is to train the diffusion model in a unified manner to model multi-task data, treating different inputs as tokens in a unified architecture, which is expected to enhance the efficiency of diverse information exploitation.

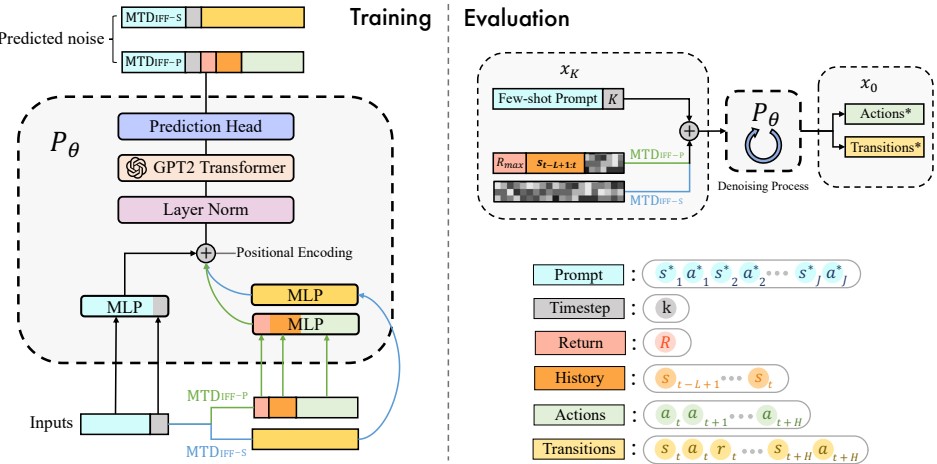

Figure 3: Model architecture of MTDIFF, which treats different inputs as tokens in a unified architecture. The two key designs are (i) the trainable GPT2 Transformer which enhances sequential modeling, and (ii) the MLPs and prediction head which enable efficient training.

As shown in Figure 3, first, different raw inputs $x$ are embedded into embeddings $h$ of the same size $\boldsymbol{d}$ via separate MLPs $f$, which can be expressed as:

$$h_P = f_P(x^{\text{prompt}}), h_{Ti} = f_{Ti}(x^{\text{timestep}}), \quad \triangleright \text{ for prompt and diffusion timestep}$$

$$h_{Tr}^s = f_{Tr}(x^{\text{transitions}}), \quad \triangleright \text{ for MTDIFF-S}$$

$$h_A^p = f_A(x^{\text{actions}}), h_H^p = f_H(x^{\text{history}}), h_R^p = f_R(x^{\text{return}}). \quad \triangleright \text{ for MTDIFF-P}$$

Then, the embeddings $h_P$ and $h_{Ti}$ are prepended as follows to formulate input tokens for MTDIFF-P and MTDIFF-S, respectively:

$$h_{\text{tokens}}^p = \text{LN}(h_{Ti} \times [h_P, h_{Ti}, h_R^p, h_H^p, h_A^p] + h_R^p + E^{\text{pos}}), h_{\text{tokens}}^s = \text{LN}(h_{Ti} \times [h_P, h_{Ti}, h_{Tr}^s] + E^{\text{pos}}),$$

where $E^{\text{pos}}$ is the positional embedding, and LN denotes layer normalization [3] for stabilizing training. In our implementation, we strengthen the condition of the stacked inputs through multiplication with the diffusion timestep $h_{Ti}$ and addition with the return $h_R^p$. GPT2 is a decoder-only transformer that incorporates a self-attention mechanism to capture dependencies between different positions in the input sequence. We employ the GPT2 architecture as a trainable backbone in MTDIFF to handle sequential inputs. It outputs an updated representation as:

$$h_{\text{out}}^p = \text{transformer}(h_{\text{tokens}}^p), \quad h_{\text{out}}^s = \text{transformer}(h_{\text{tokens}}^s).$$

Finally, given the output representation, we use a prediction head consisting of fully connected layers to predict the corresponding noise at diffusion timestep $k$. Notice that the predicted noise shares the same dimensional space as the original inputs, which differs from the representation size $\boldsymbol{d}$. This noise is used in the reverse denoising process $p_\theta$ during inference. We summarize the details of the training process, architecture and hyperparameters used in MTDIFF in Appendix A.

## 4 Related Work

**Diffusion Models in RL.** Diffusion models have emerged as a powerful family of deep generative models with a record-breaking performance in many applications across vision, language and combinatorial optimization [49, 46, 17, 31, 32]. Recent works in RL have demonstrated the capability of diffusion models to learn the multimodal distribution of offline policies [63, 43, 11] or human behaviors [8]. Other works formulate the sequential decision-making problem as a conditional generative process [2] and learn to generate the trajectories satisfying conditioned constraints. However, these works are limited to the single-task settings, while we further study the trajectory modeling and generalization problems of diffusion models in multi-task settings.

**Multi-Task RL and Few-Shot RL.** Multi-task RL aims to learn a shared policy for a diverse set of tasks. The main challenge of multi-task RL is the conflicting gradients among different tasks, and

previous online RL works address this problem via gradient surgery [74], conflict-averse learning [34], and parameter composition [70, 56]. Instead, MTDIFF addresses such a problem in an offline setting through a conditional generative process via a novel transformer architecture. Previous Decision-Transformer (DT)-based methods [47, 29, 72] which consider handling multi-task problems, mainly rely on expert trajectories and entail substantial training expenses. Scaled-QL [26] adopts separate networks for different tasks and is hard to generalize to new tasks. Instead of focusing on the performance of training tasks in multi-task RL, few-shot RL aims to improve the generalizability in novel tasks based on the learned multi-task knowledge. Nevertheless, these methods need additional context encoders [77, 79] or gradient descents in the finetuning stage [57, 58, 29]. In contrast, we use prompts for few-shot generalization without additional parameter-tuning.

**Data Augmentation for RL.** Data augmentation [13, 39] has been verified to be effective in RL. Previous methods incorporate various data augmentations (e.g. adding noise, random translation) on observations for visual-based RL [71, 28, 52, 27], which ensure the agents learn on multiple views of the same observation. Differently, we focus on data augmentation via synthesizing new experiences rather than perturbing the origin one. Recent works [76, 10] consider augmenting the observations of robotic control using a text-guided diffusion model whilst maintaining the same action, which differs from our approach that can synthesize novel action and reward labels. The recently proposed SynthER [38] is closely related to our method by generating transitions of trained tasks via a diffusion model. However, SynthER is studied in single-task settings, while we investigate whether a diffusion model can accommodate all knowledge of multi-task datasets and augment the data for novel tasks.

## 5 Experiments

In this section, we conduct extensive experiments to answer the following questions: (1) How does MTDIFF-P compare to other offline and online baselines in the multi-task regime? (2) Does MTDIFF-S synthesize high-fidelity data and bring policy improvement? (3) How is MTDIFF-S compared with other augmentation methods for single-task RL? (4) Does the synthetic data of MTDIFF-S match the original data distribution? (5) Can both MTDIFF-P and MTDIFF-S generalize to unseen tasks?

### 5.1 Environments and Baselines

**Meta-World Tasks**  The Meta-World benchmark [73] contains 50 qualitatively-distinct manipulation tasks. The tasks share similar dynamics and require a Sawyer robot to interact with various objects with different shapes, joints, and connectivity. In this setup, the state space and reward functions of different tasks are different since the robot is manipulating different objects with different objectives. At each timestep, the Sawyer robot receives a 4-dimensional fine-grained action, representing the 3D position movements of the end effector and the variation of gripper openness. The original Meta-World environment is configured with a fixed goal, which is more restrictive and less realistic in robotic learning. Following recent works [70, 56], we extend all the tasks to a random-goal setting and refer to it as MT50-rand. We use the average success rate of all tasks as the evaluation metric.

By training a SAC [18] agent for each task in isolation, we utilize the experience collected in the replay buffer as our offline dataset. Similar to [26], we consider two different dataset compositions: (i) **Near-optimal** dataset consisting of the experience (100M transitions) from random to expert (convergence) in SAC-Replay, and (ii) **Sub-optimal** dataset consisting of the initial 50% of the trajectories (50M transitions) from the replay buffer for each task, where the proportion of expert data decreases a lot. We summarize more details about the dataset in Appendix G.

**Maze2D Tasks**  Maze2D [15] is a navigation task that requires an agent to traverse from a randomly designated location to a fixed goal in the 2D map. The reward is 1 if succeed and 0 otherwise. Maze2D can evaluate the ability of RL algorithms to stitch together previously collected sub-trajectories, which helps the agent find the shortest path to evaluation goals. We use the agent's scores as the evaluation metric. The offline dataset is collected by selecting random goal locations and using a planner to generate sequences of waypoints by following a PD controller.

**Baselines**  We compare our proposed MTDIFF (MTDIFF-P and MTDIFF-S) with the following baselines. Each baseline has the same batch size and training steps as MTDIFF. For **MTDIFF-P**, we

have following baselines: (i) **PromptDT.** PromptDT [69] built on Decision-Transformer (DT) [9] aims to learn from multi-task data and generalize the policy to unseen tasks. PromptDT generates actions based on the trajectory prompts and reward-to-go. We use the same GPT2-network as in MTDIFF-P. The main difference between our method and PromptDT is that we employ diffusion models for generative planning. (ii) **MTDT.** We extend the DT architecture [9] to learn from multi-task data. Specifically, MTDT concatenates an embedding $z$ and a state $s$ as the input tokens, where $z$ is the encoding of task ID. In evaluation, the reward-to-go and task ID are fed into the Transformer to provide task-specific information. MTDT also uses the same GPT2-network as in MTDIFF-P. Compared to MTDT, our model incorporates prompt and diffusion framework to learn from the multi-task data. (iii) **MTCQL.** Following scaled-QL [26], we extend CQL [25] with multi-head critic networks and a task-ID conditioned actor for multi-task policy learning. (iv) **MTIQL.** We extend IQL [24] for multi-task learning using a similar revision of MTCQL. The TD-based baselines (i.e., MTCQL and MTIQL) are used to demonstrate the effectiveness of conditional generative modeling for multi-task planning. (v) **MTBC.** We extend Behavior cloning (BC) to multi-task offline policy learning via network scaling and a task-ID conditioned actor that is similar to MTCQL and MTIQL.

As for **MTDIFF-S**, we compare it with two baselines that perform direct data augmentation in offline RL. (i) **RAD.** We adopt the random amplitude scaling [28] that multiplies a random variable to states, i.e., $s' = s \times z$, where $z \sim \text{Uniform}[\alpha, \beta]$. This augmentation technique has been verified to be effective for state-based RL. (ii) **S4RL.** We adopt the adversarial state training [52] by taking gradients with respect to the value function to obtain a new state, i.e. $s' \leftarrow s + \epsilon \nabla_s \mathbb{J}_Q(\pi(s))$, where $\mathbb{J}_Q$ is the policy evaluation update performed via a $Q$ function, and $\epsilon$ is the size of gradient steps. We summarize the details of all the baselines in Appendix B.

## 5.2 Result Comparison for Planning

**How does MTDIFF-P compare to baselines in the multi-task regime?** For a fair comparison, we add MTDIFF-P-ONEHOT as a variant of MTDIFF-P by replacing the prompt with a one-hot task-ID, which is used in the baselines except for PromptDT. The first action generated by MTDIFF-P is used to interact with the environment. According to Tab. 1, we have the following key observations. (i) Our method achieves better performance than baselines in both near-optimal and sub-optimal settings. For near-optimal datasets, MTDIFF-P and MTDIFF-P-ONEHOT achieve about 60% success rate, significantly outperforming other methods and performing comparably with MTBC. However, the performance of MTBC decreases a lot in sub-optimal datasets. BC is hard to handle the conflict behaviors in experi-

Table 1: Average success rate across 3 seeds on Meta-World-V2 MT50 with random goals (MT50-rand). Each task is evaluated for 50 episodes.

| Methods | Near-optimal | Sub-optimal |
|---|---|---|
| **CARE** [53] (Online) | $50.8 \pm 1.0$ | — |
| **PaCo** [56] (Online) | $57.3 \pm 1.3$ | — |
| **MTDT** | $20.99 \pm 2.66$ | $20.63 \pm 2.21$ |
| **PromptDT** | $45.68 \pm 1.84$ | $39.76 \pm 2.79$ |
| **MTBC** | $60.39 \pm 0.86$ | $34.53 \pm 1.25$ |
| **MTCQL** | — | — |
| **MTIQL** | $56.21 \pm 1.39$ | $43.28 \pm 0.90$ |
| **MTDIFF-p (ours)** | $59.53 \pm 1.12$ | $\mathbf{48.67 \pm 1.32}$ |
| **MTDIFF-P-ONEHOT (ours)** | $\mathbf{61.32 \pm 0.89}$ | $\mathbf{48.94 \pm 0.95}$ |

ences sampled by a mixture of policies with different returns, which has also been verified in previous offline imitation methods [35, 68]. In contrast, both MTDIFF-P and MTDIFF-ONEHOT perform the best in sub-optimal datasets. (ii) We compare MTDIFF-P with two SOTA multi-task online RL methods, CARE [53] and PaCo [56], which are trained for 100M steps in MT50-rand. MTDIFF-P outperforms both of them given the near-optimal dataset, demonstrating the potential of solving the multi-task RL problems in an offline setting. (iii) MTDT is limited in distinguishing different tasks with task ID while PromptDT performs better, which demonstrates the effect of prompts in multi-task settings. (iv) As for TD-based baselines, we find MTCQL almost fails while MTIQL performs well. We hypothesize that since MTCQL penalizes the OOD actions for each task, it will hinder other tasks' learning since different tasks can choose remarkably different actions when facing similar states. In contrast, IQL learns a value function without querying the values of OOD actions.

We remark that MTDIFF-P based on GPT network outperforms that with U-Net in similar model size, and detailed results are given in Appendix C. Overall, MTDIFF-P is an effective planner in multi-task settings including sub-optimal datasets where we must stitch useful segments of suboptimal trajectories, and near-optimal datasets where we mimic the best behaviors. Meanwhile, we argue that although MTDIFF-P-ONEHOT performs well, it cannot generalize to unseen tasks without prompts.

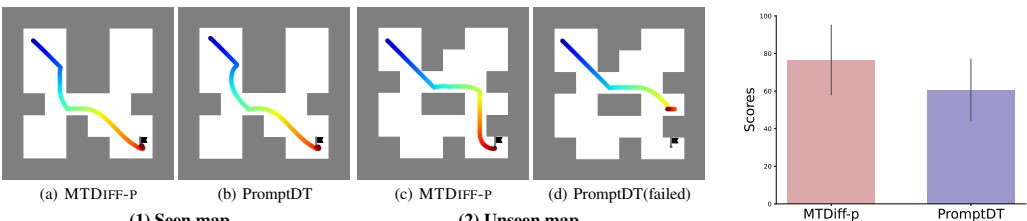

| (a) MTDIFF-P | (b) PromptDT | (c) MTDIFF-P | (d) PromptDT(failed) |
|:---:|:---:|:---:|:---:|
| **(1) Seen map** | | **(2) Unseen map** | |

Figure 4: Seen and unseen maps of Maze2D with long planning path. Goal position is denoted as ▪.

Figure 5: Average scores obtained in 8 Maze2D maps.

**Does MTDIFF-P generalize to unseen tasks?** We further carry out experiments on Maze2D to evaluate the generalizability of MTDIFF-P. We select PromptDT as our baseline, as it has demonstrated both competitive performances on training tasks and adaptation ability for unseen tasks [69]. We use 8 different maps for training and one new map for adaptation evaluation. The setup details are given in Appendix D. We evaluate these two methods on both seen maps and an unseen map. The average scores obtained in 8 training maps are referred to Figure 5. To further illustrate the advantages of our method compared to PromptDT, we select one difficult training map and the unseen map for visualization, as shown in Figure 4. According to the visualized path, we find that (1) for seen maps in training, MTDIFF-P generates a shorter and smoother path, and (2) for unseen maps, PromptDT fails to obtain a reasonable path while MTDIFF-P succeed, which verifies that MTDIFF-P can perform few-shot adaptation based on trajectory prompts and the designed architecture.

## 5.3 Results for Augmentation via Data Synthesis

**Does MTDIFF-S synthesize high-fidelity data and bring policy improvement?** We train MTDIFF-S on near-optimal datasets from 45 tasks to evaluate its generalizability. We select 3 training tasks and 3 unseen tasks, and measure the policy improvement of offline RL training (i.e., TD3-BC [16]) with data augmentation. For each evaluated task, MTDIFF-S synthesizes 2M transitions to expand the original 1M dataset. From the results summarized in Table 2, MTDIFF-S can boost the offline performance for all tasks and significantly increases performance by about 180%, 131%, and 161% for *box-close*, *hand-insert*, and *coffee-push*, respectively.

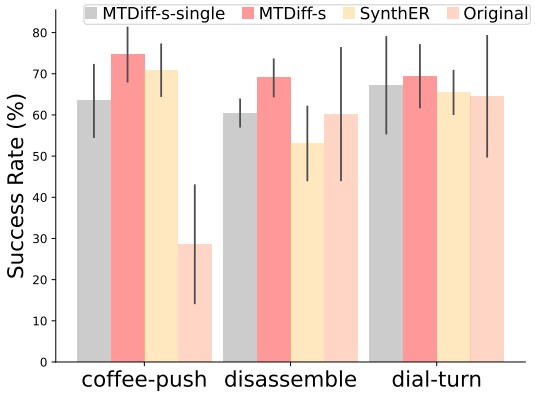

Figure 6: Results of multi-task and single- task augmentation

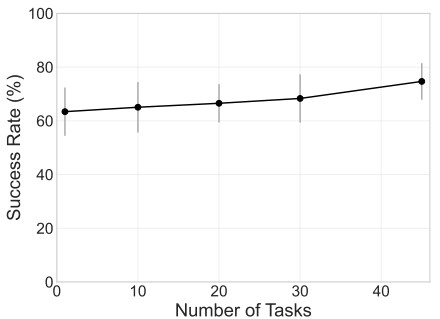

Figure 7: Average success rate across 3 seeds on *coffee-push*. Dataset is augmented by MTDIFF-S trained under different number of tasks.

**How does MTDiff-s perform and benefit from multi-task training?** From Table 2, we find MTDIFF-S achieves superior policy improvement in seen tasks compared with previous SOTA augmentation methods (i.e., S4RL and RAD) that are developed in single-task RL. We hypothesize that, by absorbing vast knowledge of multi-task data in training, MTDIFF-S can perform implicit data sharing [75] by integrating other tasks' knowledge into data synthesis of the current task. To verify this hypothesis, we select three tasks (i.e., *coffee-push*, *disassemble* and *dial-turn*) to re-train MTDIFF-S on the corresponding single-task dataset. We denote this variant as MTDIFF-S-SINGLE. We implement SynthER [38] to further confirm the benefits of multi-task training. SynthER also utilizes diffusion models to enhance offline datasets for policy improvement, but it doesn't take into

account the aspect of learning from multi-task data. We observe that MTDIFF-S outperforms both two methods, as shown in Figure 6. What's more, we re-train MTDIFF-S on 10, 20, and 30 tasks respectively, in order to obtain the relationship between the performance gains and the number of training tasks. Our findings, as outlined in Fig. 7, provide compelling evidence in support of our hypothesis: MTDIFF-S exhibits progressively superior data synthesis performance with increasing task diversity.

**Does MTDIFF-S generalize to unseen tasks?** We answer this question by conducting offline RL training on the augmented datasets of 3 unseen tasks. According to Table 2, MTDIFF-S is well-generalized and obtains significant improvement compared to the success rate of original datasets. MTDIFF-S boost the policy performance by 131%, 180% and 32% for *hand-insert*, *box-close* and *bin-picking*, respectively. We remark that S4RL performs the best on the two unseen tasks, i.e., *box-close* and *bin-picking*, since it utilizes the entire datasets to train $Q$-functions and obtains the augmented states. Nevertheless, we use much less information (i.e., a single trajectory as prompts) for augmentation.

Table 2: Average success rate across 3 seeds on Meta-World-V2 single task with random goals. Each selected task is evaluated for 500 episodes.

| Tasks | | Original | S4RL | RAD | MTDIFF-S(OURS) |
|---|---|---|---|---|---|
| **Unseen Tasks** | **box-close** | $23.46 \pm 7.11$ | $\mathbf{73.13 \pm 3.51}$ (↑ 211%) | $71.20 \pm 4.84$ (↑ 203%) | $65.73 \pm 8.36$ (↑ 180%) |
| | **hand-insert** | $30.60 \pm 9.77$ | $60.20 \pm 1.57$ (↑ 96%) | $43.79 \pm 3.44$ (↑ 43%) | $\mathbf{70.87 \pm 3.59}$ (↑ 131%) |
| | **bin-picking** | $42.13 \pm 14.33$ | $\mathbf{72.20 \pm 4.17}$(↑ 71%) | $43.27 \pm 4.38$(↑ 2%) | $55.73 \pm 7.63$(↑ 32%) |
| **Seen Tasks** | **sweep-into** | $91.8 \pm 1.14$ | $90.53 \pm 3.52$(↓ 1%) | $88.06 \pm 9.86$(↓ 4%) | $\mathbf{92.87 \pm 1.11}$(↑ 1%) |
| | **coffee-push** | $28.60 \pm 14.55$ | $28.73 \pm 8.44$(↑ 0.4%) | $33.19 \pm 2.86$(↑ 16%) | $\mathbf{74.67 \pm 6.79}$(↑ 161%) |
| | **disassemble** | $60.20 \pm 16.29$ | $52.20 \pm 5.68$(↓ 12%) | $60.93 \pm 20.80$(↑ 1%) | $\mathbf{69.00 \pm 4.72}$(↑ 14.6%) |

**Does the synthetic data of MTDIFF-S match the original data distribution?** We select 4 tasks and use T-SNE [61] to visualize the distribution of original data and synthetic data. We find the synthetic data overlap and expand the original data distribution while also keeping consistency with the underlying MDP. The visualization results and further analyses are given in Appendix E.

## 6  Conclusion

We propose MTDIFF, a diffusion-based effective planner and data synthesizer for multi-task RL. With the trajectory prompt and unified GPT-based architecture, MTDIFF can model multi-task data and generalize to unseen tasks. We show that in the MT50-rand benchmark containing fine-grained manipulation tasks, MTDIFF-P generates desirable behavior for each task via few-shot prompts. By compressing multi-task knowledge in a single model, we demonstrate that MTDIFF-S greatly boosts policy performance by augmenting original offline datasets. In future research, we aim to develop a practical multi-task algorithm for real robots to trade off the sample speed and generative quality. We further discuss the limitations and broader impacts of MTDIFF in Appendix F.

## Acknowledgments

This work is supported by the National Natural Science Foundation of China (Grant No.62306242&62076161), the National Key R&D Program of China (Grant No.2022ZD0160100), Shanghai Municipal Science and Technology Major Project (2021SHZDZX0102) and Shanghai Artificial Intelligence Laboratory.

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

# A  The Details of MTDIFF

In this section, we give the pseudocodes of MTDIFF-P and MTDIFF-S in Alg. 1 and Alg. 2, respectively. Then we describe various details of the training process, architecture and hyperparameters:

- We set the per-task batch size as 8, so the total batch size is 400. We train our model using Adam optimizer [23] with $2e^{-4}$ learning rate for $2e^6$ train steps.

- We train MTDIFF on NVIDIA GeForce RTX 3080 for around 50 hours.

- We represent the noise model as the transformer-based architecture described in Section 3.3. MLP $f_P$ which processes prompt is a 3-layered MLP (prepended by a layer norm [3] and with Mish activation). MLP $f_{Ti}$ which processes diffusion timestep $f_R$ is a 2-layered MLP (prepended by a Sinusoidal embedding and with Mish activation). $f_R$ which processes conditioned Return and $f_H$ which processes state history are 3-layered MLPs with Mish activation. $f_A$ which processes actions, $f_{TR}$ which process transitions and prediction head are 2-layered MLPs with Mish activation. The GPT2 transformer is configured as 6 hidden layers and 4 attention heads. The code of GPT2 is borrowed from `https://github.com/kzl/decision-transformer`.

- We choose the probability $p$ of removing the conditioning information to be 0.25.

- In MTDIFF-P, we choose the state history length $L = 2$ for Meta-World and $L = 5$ for Maze2D.

- We choose the trajectory prompt length $J = 20$.

- We use $K = 200$ for diffusion steps.

- We set guidance scale $\alpha = 1.2$ for extracting near-optimal behavior.

- We choose $\beta = 0.5$ for low temperature sampling.

---

**Algorithm 1** MTDIFF-P Training and Evaluation

---
*# Training Process*
**Initialize**: training tasks $\mathcal{T}^{train}$, training iterations $N$, multi-task dataset $\mathcal{D}$, per-task batch size $M$, multi-task trajectory prompts $Z$

1: **for** $n = 1$ **to** $N$ **do**
2:     **for** Each task $\mathcal{T}_i \in \mathcal{T}^{train}$ **do**
3:         Sample action sequences $\boldsymbol{x}_0^p(\tau_i)$ of length $H$ and corresponding state history $S_i^{\mathrm{prev}}$ of length $L$ from $\mathcal{D}_i$ with batch size $M$
4:         Compute normalized return $R(\tau_i)$ under $\tau_i$
5:         Sample trajectory prompts $\tau_i^*$ of length $J$ from $Z_i$ with batch size $M$
6:     **end for**
7:     Get a batch $\mathcal{B} = \{\boldsymbol{x}_0^p(\tau_i), \tau_i^*, S_i^{\mathrm{prev}}, R(\tau_i)\}_{i=1}^{|\mathcal{T}^{train}|}$
8:     Randomly sample a diffusion timestep $k \sim \mathcal{U}(1, K)$ and obtain noisy sequences $\boldsymbol{x}_k^p(\tau_i)$
9:     Omit the $R(\tau)$ condition with probability $\beta \sim \mathrm{Bern}(p)$
10:    Compute $\mathcal{L}^p(\theta)$ and update MTDIFF-P model
11: **end for**

*# Evaluation Process*
1: Given a task, reset the environment and set desired return $R_{\max}(\tau)$
2: Obtain the initial state history $h_0$, few-shot prompts $Z$
3: Set low-temperature sampling scale $\beta$, classifier-free guidance scale $\alpha$
4: **for** $t = 0$ **to** $t_{\max}$ **do**
5:     Initialize $\boldsymbol{x}_K^p(\tau) \sim \mathcal{N}(\boldsymbol{0}, \beta\boldsymbol{I})$
6:     Sample $\tau^* \sim Z$, and formulate $\boldsymbol{y}'(\tau) = [h_t, \tau^*]$
7:     **for** $k = K$ **to** 1 **do**
8:         $\hat{\epsilon} = \epsilon_\theta\big(\boldsymbol{x}_k^p(\tau), \boldsymbol{y}'(\tau), \varnothing, k\big) + \alpha\big(\epsilon_\theta(\boldsymbol{x}_k^p(\tau), \boldsymbol{y}'(\tau), R_{\max}(\tau), k) - \epsilon_\theta(\boldsymbol{x}_k^p(\tau), \boldsymbol{y}'(\tau), \varnothing, k)\big)$
9:         $(\mu_{k-1}, \Sigma_{k-1}) \leftarrow \mathrm{Denoise}(\boldsymbol{x}_k^p(\tau), \hat{\epsilon})$
10:       $\boldsymbol{x}_{k-1}^p(\tau) \sim \mathcal{N}(\mu_{k-1}, \beta\Sigma_{k-1})$
11:     **end for**
12:    Execute the first action from $\boldsymbol{x}_0^p(\tau)$ as the current action to interact with the environment
13:    Obtain the next state, and update $h_t$
14: **end for**

---

---

**Algorithm 2** MTDIFF-S Training and Data Synthesis

---

*# Training Process*

**Initialize**: training tasks $\mathcal{T}^{train}$, training iterations $N$, multi-task dataset $\mathcal{D}$, per-task batch size $M$, multi-task trajectory prompts $Z$

1: **for** $n = 1$ **to** $N$ **do**
2:     **for** Each task $\mathcal{T}_i \in \mathcal{T}^{train}$ **do**
3:         Sample transition sequences $\boldsymbol{x}_0^s(\tau_i)$ of length $H$ from $\mathcal{D}_i$ with batch size $M$
4:         Sample trajectory prompts $\tau_i^*$ of length $J$ from $Z_i$ with batch size $M$
5:     **end for**
6:     Get a batch $\mathcal{B} = \{\boldsymbol{x}_0^s(\tau_i), \tau_i^*\}_{i=1}^{|\mathcal{T}^{train}|}$
7:     Randomly sample a diffusion timestep $k \sim \mathcal{U}(1, K)$ and obtain noisy sequences $\boldsymbol{x}_k^s(\tau_i)$
8:     Compute $\mathcal{L}^s(\theta)$ and update MTDIFF-S model
9: **end for**

*# Data Synthesis Process*

**Initialize**: synthetic dataset $\mathcal{D} = \varnothing$, synthesizing times $M$

1: Given a task, obtain few-shot prompts $Z$
2: **for** $m = 1$ **to** $M$ **do**
3:     Initialize $\boldsymbol{x}_K^s(\tau) \sim \mathcal{N}(\boldsymbol{0}, \boldsymbol{I})$
4:     Sample $\tau^* \sim Z$, and formulate $\boldsymbol{y}_k^s(\tau) = [\tau^*]$
5:     **for** $k = K$ **to** 1 **do**
6:         $\hat{\epsilon} = \epsilon_\theta\big(\boldsymbol{x}_k^s(\tau), \boldsymbol{y}^s(\tau), k\big)$
7:         $(\mu_{k-1}, \Sigma_{k-1}) \leftarrow \text{Denoise}(\boldsymbol{x}_k^s(\tau), \hat{\epsilon})$
8:         $\boldsymbol{x}_{k-1}^s(\tau) \sim \mathcal{N}(\mu_{k-1}, \Sigma_{k-1})$
9:     **end for**
10:    Update $\mathcal{D} = \mathcal{D} \cup \boldsymbol{x}_0^s(\tau)$
11: **end for**

---

# B  The Details of Baselines

In this section, we describe the implementation details of the baselines:

- **PromptDT** uses the same prompts and GPT2 transformer in MTDIFF-P for taining. We borrow the code from `https://github.com/mxu34/prompt-dt` for implementation.

- **MTDT** embeds the tskaID which indicates the task to a embedding $z$ with size 12, then $z$ is concatenated with the raw state. With such conditioning, we broaden the original state space to equip DT with the ability to identify tasks in this multi-task setting. We keep other hyperparameters and implementation details the same as the official version `https://github.com/kzl/decision-transformer/`.

- **MTIQL** uses a multi-head critic network to predict the $Q$ value for each task, and each head is parameterized with a 3-layered MLP (with Mish activation). The actor-network is parameterized with a 3-layered MLP (with Mish activation). During training and inference, the scalar taskID is embedded via 3-layered MLP (with Mish activation) into latent variable $z$, and the input of the actor becomes the concatenation of the original state and $z$. We build MTIQL based on the code `https://github.com/tinkoff-ai/CORL` [59].

- **MTCQL** is applied with a similar revision in MTIQL. The main difference is that MTCQL is based on CQL [25] algorithm instead of the IQL algorithm [24]. We build MTCQL based on the code `https://github.com/tinkoff-ai/CORL` [59].

- **MTBC** uses a similar taskID-cognitional actor in MTIQL and MTCQL. For training and inference, the scalar taskID is embedded via a 3-layered MLP (with Mish activation) into latent variable $z$, and the input of the actor becomes the concatenation of the original state and $z$. The actor is parameterized with a 3-layered MLP and outputs predicted actions.

- **RAD** adopts the random amplitude scaling [28] that multiplies a random variable to states during training, i.e., $s' = s \times z$, where $z \sim \text{Uniform}[\alpha, \beta]$. We choose $\alpha = 0.8$ and $\beta = 1.2$.

- **S4RL** adopts the adversarial state training [52] by taking gradients with respect to the value function to obtain a new state, i.e. $s' \leftarrow s + \epsilon\nabla_s\mathbb{J}_Q(\pi(s))$, where $\mathbb{J}_Q$ is the policy evaluation update performed via a $Q$ function, and $\epsilon$ is the size of gradient steps. We choose $\epsilon = 0.01$.

## C  Ablation Study on Model Architecture

The architecture described in §3.3 handles input types of different modalities as tokens that share similar formats, actively capturing interactions between modalities. The incorporation of a transformer is also helpful for sequential modeling. To ablate the effectiveness of our architecture design, we train MTDIFF-P using U-Net with a similar model size to ours on the near-optimal dataset. We use a Temporal Convolutional Network (TCN) [4] to encode the prompt into an embedding $z_p$, and inject it in the U-Net layers as a condition. We follow the conditional approach in [2] and borrow the code for temporal U-Net from `https://github.com/jannerm/diffuser` [21]. The results summarized in Table 3 show that our model architecture outperforms U-Net to learn from multi-task datasets.

Table 3: Average success rate across 3 different seeds of MTDIFF-P and MTDIFF-P (U-Net) on MT50-rand.

| Methods | Success rate on near-optimal dataset (%) | Success rate on sub-optimal dataset (%) |
|---|---|---|
| MTDIFF-P | **59.53 $\pm$ 1.12** | **48.67 $\pm$ 1.32** |
| MTDIFF-P (U-Net) | 55.67 $\pm$ 1.27 | 47.42 $\pm$ 0.74 |

## D  Environmental Details of Maze2D

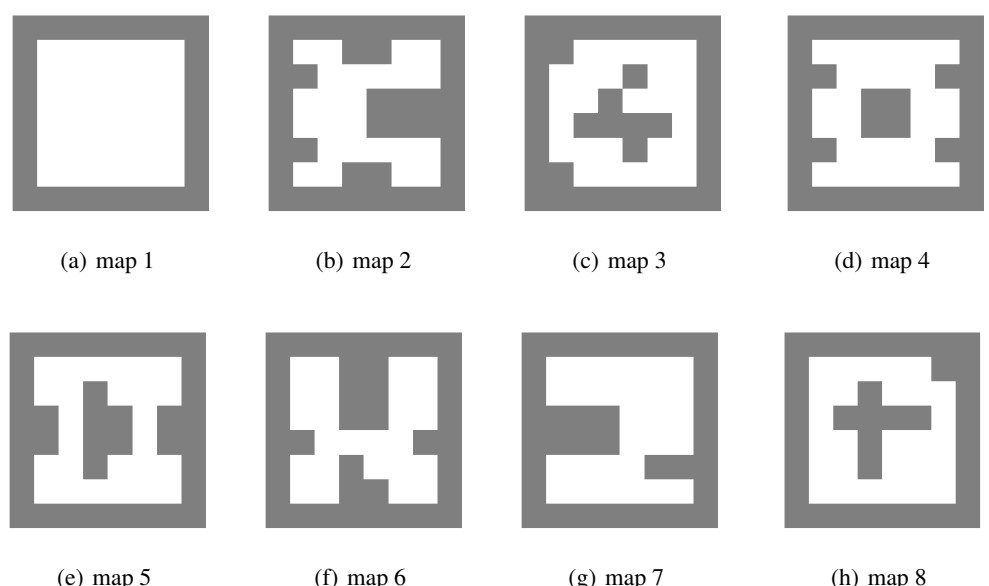

(a) map 1    (b) map 2    (c) map 3    (d) map 4

(e) map 5    (f) map 6    (g) map 7    (h) map 8

Figure 8: 2D visualization of eight training maps designed in Maze2D.

We design eight different training maps for multi-task training, which are shown at Fig. 8. Different tasks have different reward functions and transition functions. For generalizability evaluation, we have designed one new unseen map. Although in Fig. 4, MTDIFF is able to generalize on new maps while PromptDT fails, we should acknowledged that MTDIFF may fail at some difficult unseen cases, as shown in Fig. 9. The reasons may lie in 2 folds: One is the inherent difficulty of the case itself, and the other is that the case's deviation from the distribution of the training cases surpasses the upper threshold of generalizability of MTDIFF. We also provide another map where MTDIFF succeeds while PromptDT fails in Fig. 10.

We collect 35k episodes together to train our model and PromptDT. The episodic length is set as 600 for training and 200 for evaluation. After training with 512 batch size for 2e5 gradient steps, we evaluate these methods on seen and unseen maps.

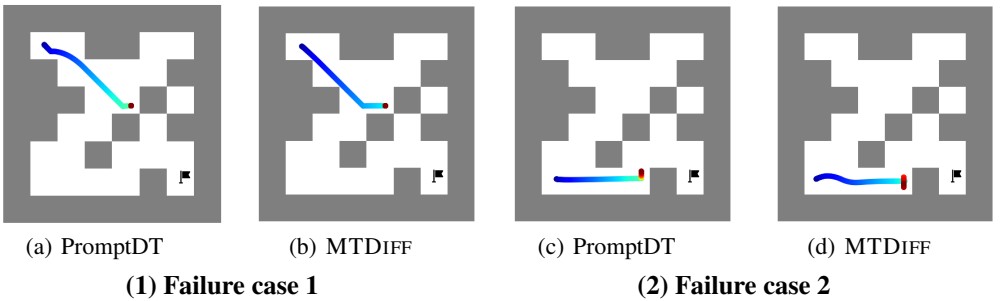

(a) PromptDT      (b) MTDIFF      (c) PromptDT      (d) MTDIFF

**(1) Failure case 1**          **(2) Failure case 2**

Figure 9: 2D visualization of 2 difficult cases where both PromptDT and MTDIFF both fail. These 2 cases are both unseen during training. Goal position is denoted as ⚑.

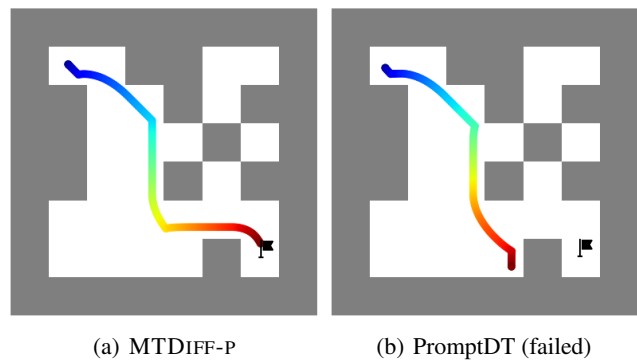

(a) MTDIFF-P        (b) PromptDT (failed)

Figure 10: Unseen maps of Maze2D with long planning path. MTDIFF-P reach the goal while PromptDT fails. Goal position is denoted as ⚑.

## E    Data Analysis

### E.1    Distribution Visualization

We find the synthetic data is high-fidelity, covering or even broadening the original data distribution, which makes the offline RL method performs better in the augmented dataset. The distribution visualization is shown in Fig. 11.

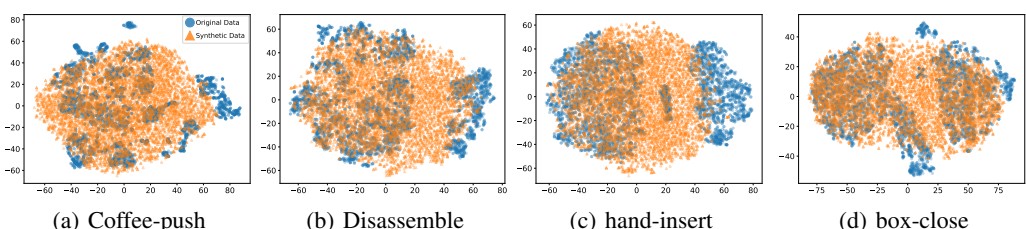

(a) Coffee-push     (b) Disassemble     (c) hand-insert     (d) box-close

Figure 11: 2D visualization of sampled synthetic data and original data via T-SNE [61]. The data is sampled from tasks coffee-push, disassemble, hand-insert and box-close respectively.

### E.2 Statistical Analysis

Following SynthER [38], we measure the Dynamics Error (MSE between the augmented next state and true next state), and L2 Distance from Dataset (minimum L2 distance of each datapoint from the dataset) for the augmented data of each method (i.e., MTDIFF-S, S4RL and RAD), as shown in Table 4. Since S4RL performs data augmentation by adding data within the $\epsilon$-ball of the original data points, it has the smallest dynamics error with a small $\epsilon$. RAD performs random amplitude scaling and causes the largest dynamics error. We remark that S4RL performs local data augmentation around the original data and can be limited in expanding the data coverage of offline datasets. In contrast, our method generates data via diffusion model without explicit constraints to the original data points, which also has small dynamics error and significantly improves the data coverage, benefiting the offline RL training.

Table 4: Comparing L2 distance from the training dataset and dynamics error under each method.

| Methods | Dynamics Error | L2 Distance from Dataset |
|---------|----------------|--------------------------|
| MTDIFF-S | 0.0174 | 0.4552 |
| S4RL | 0.0001 | 0.4024 |
| RAD | 0.0641 | 0.4617 |

## F Limitations and Discussions

In this section, we will discuss the limitations and broader impacts of our proposed method MTDIFF.

**Limitation.** Diffusion models are bottlenecked by their slow sampling speed, which caps the potential of MTDIFF for real-time control. How to trade off the sampling speed and generative quality remains to be a crucial research topic. For a concrete example in MetaWorld, it takes on average 1.9s in wall-clock time to generate one action sequence for planning (hardware being a 3090 GPU). We can improve the inference speed by leveraging a recent sampler called DPM-solver [37, 36] to decrease the diffusion steps required to $0.2\times$ without any loss in performance, and using a larger batch size (leveraging the parallel computing power of GPUs) to evaluate multiple environments at once. Thus the evaluation run-time roughly matches the run-time of non-diffusion algorithms (diffusion step is 1). In addition, consistency models [55] are recently proposed to support one-step and few-step generation, while the upper performance of what such models can achieve is still vague.

**Broader Impacts.** As far as we know, MTDIFF is the first proposed diffusion-based approach for multi-task reinforcement learning. It could be applied to multi-task decision-making, and also could be used to synthesize more data to boost policy improvement. MTDIFF provides a solution for achieving generalization in reinforcement learning.

## G Dataset collection

**Meta-World.** We train Soft Actor-Critic (SAC) [18] policy in isolation for each task from scratch until convergence. Then we collect 1M transitions from the SAC replay buffer for each task, consisting of recording samples in the replay buffer observed during training until the policy reaches the convergence of performance. For this benchmark, we have two different dataset compositions:

- **Near-optimal** dataset consisting of the experience (100M transitions) from random to expert (convergence) in SAC-Replay.

- **Sub-optimal** dataset consisting of the initial 50% of the trajectories (50M transitions) of the near-optimal dataset for each task, where the proportion of expert data decreases a lot.

To visualize the optimality of each dataset clearly, we plot the univariate distribution of return in each kind of dataset in Fig. 12. Our dataset is available at `https://bit.ly/3MWf40w`.

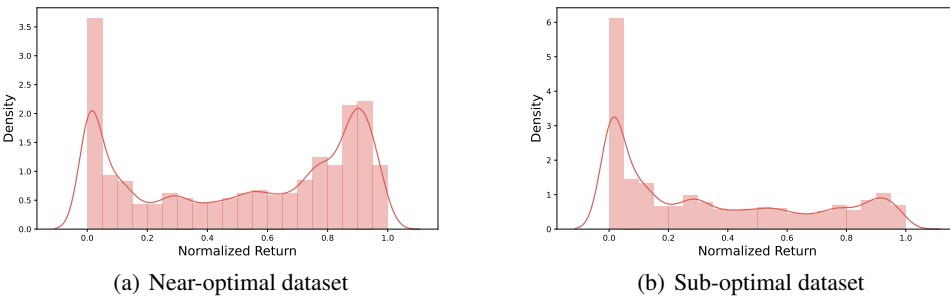

(a) Near-optimal dataset  (b) Sub-optimal dataset

Figure 12: Density visualization of the normalized return in the dataset.

**Maze2D**   The offline dataset is collected by selecting random goal locations and using a planner to generate sequences of waypoints by following a PD controller. We borrow the code from `https://github.com/Farama-Foundation/D4RL` [15] to generate datasets for 8 training maps. We collect 35k episodes in total.

## H   Differences Between PromptDT and MTDIFF-P

The remarkable superiority of MTDIFF-P over PromptDT emerges from our elegant incorporation of transformer architecture and trajectory prompt within the diffusion model framework, effectively modeling the multi-task trajectory distribution. PromptDT is built on Decision Transformer and it is trained in an autoregressive manner, which is limited to predicting actions step by step. However, MTDIFF-P leverages the potency of sequence modeling, empowering it to perform trajectory generation adeptly. MTDIFF-P has demonstrated SOTA performance in both multi-task decision-making and data synthesis empirical experiments, while PromptDT fails to contribute to data synthesis. Technically, MTDIFF-P extends Decision Diffuser [2] into the multi-task scenario, utilizing classifier-free guidance for generative planning to yield high expected returns. To further verify our claim, we train our model on the publicly available PromptDT datasets [69], i.e., Cheetah-vel and Ant-dir. These chosen environments have been judiciously selected due to their inherent diversity of tasks, serving as a robust test to validate the capability of multi-task learning. We report the scores (mean and std for 3 seeds) in Table 5.

Table 5: Average scores obtained by MTDIFF-P and PromptDT across 3 seeds. We observed that MTDIFF-P outperforms PromptDT largely, demonstrating its high efficacy and potency.

| Methods | Cheetah-vel | Ant-dir |
|---------|-------------|---------|
| MTDIFF-P | $-29.09 \pm 0.31$ | $602.17 \pm 1.68$ |
| PromptDT | $-34.43 \pm 2.33$ | $409.81 \pm 9.69$ |

## I   Single-Task Performance

We train one MTDIFF-P model on MT50-rand and evaluate the performance for each task for 50 episodes. We report the average evaluated return in Table 6.

Table 6: Evaluated return of MTDIFF-P for each task in MT50-rand. We report the mean and standard deviation for 50 episodes for each task.

| Tasks | Return on near-optimal dataset | Return on sub-optimal dataset |
|---|---|---|
| basketball-v2 | $2735.7 \pm 1927.7$ | $2762.7 \pm 1928.8$ |
| bin-picking-v2 | $733.7 \pm 1211.8$ | $59.6 \pm 32.3$ |
| button-press-topdown-v2 | $1491.6 \pm 390.9$ | $1395.8 \pm 332.6$ |
| button-press-v2 | $2419.2 \pm 413.7$ | $2730.4 \pm 514.3$ |
| button-press-wall-v2 | $3474.7 \pm 887.6$ | $2613.1 \pm 906.8$ |
| coffee-button-v2 | $3157.9 \pm 1274.3$ | $1649.4 \pm 869.5$ |
| coffee-pull-v2 | $437.5 \pm 597.3$ | $98.8 \pm 115.4$ |
| coffee-push-v2 | $463.7 \pm 729.1$ | $71.3 \pm 55.7$ |
| dial-turn-v2 | $2848.1 \pm 996.3$ | $2244.5 \pm 881.5$ |
| disassemble-v2 | $369.9 \pm 265.4$ | $372.3 \pm 623.2$ |
| door-close-v2 | $4325.2 \pm 377.6$ | $4270.4 \pm 460.1$ |
| door-lock-v2 | $3215.1 \pm 857.5$ | $3082.4 \pm 1041.4$ |
| door-open-v2 | $3458.1 \pm 960.5$ | $2457.0 \pm 833.0$ |
| door-unlock-v2 | $2082.6 \pm 1568.4$ | $3078.8 \pm 1158.1$ |
| hand-insert-v2 | $927.9 \pm 1527.6$ | $288.8 \pm 932.1$ |
| drawer-close-v2 | $4824.5 \pm 8.8$ | $4825.3 \pm 22.3$ |
| drawer-open-v2 | $3530.4 \pm 1512.6$ | $2218.8 \pm 529.3$ |
| faucet-open-v2 | $3877.4 \pm 598.1$ | $4245.6 \pm 575.2$ |
| faucet-close-v2 | $4167.1 \pm 418.8$ | $4624.1 \pm 107.5$ |
| handle-press-side-v2 | $3423.3 \pm 1075.6$ | $3995.8 \pm 1130.0$ |
| handle-press-v2 | $3397.9 \pm 1752.1$ | $3125.4 \pm 1724.7$ |
| handle-pull-side-v2 | $3141.5 \pm 1775.4$ | $1474.2 \pm 1392.9$ |
| handle-pull-v2 | $2845.7 \pm 1970.6$ | $2856.8 \pm 1699.8$ |
| lever-pull-v2 | $3383.5 \pm 1299.2$ | $3125.6 \pm 1433.1$ |
| peg-insert-side-v2 | $915.1 \pm 1624.8$ | $497.8 \pm 1015.3$ |
| pick-place-wall-v2 | $649.8 \pm 1288.6$ | $288.2 \pm 988.1$ |
| pick-out-of-hole-v2 | $1792.8 \pm 1959.9$ | $700.5 \pm 1190.3$ |
| reach-v2 | $4144.2 \pm 645.4$ | $966.4 \pm 842.9$ |
| push-back-v2 | $97.2 \pm 611.3$ | $566.3 \pm 1042.9$ |
| push-v2 | $142.0 \pm 335.7$ | $64.6 \pm 98.2$ |
| pick-place-v2 | $166.0 \pm 759.2$ | $7.3 \pm 4.8$ |
| plate-slide-v2 | $4096.4 \pm 1202.5$ | $4306.3 \pm 495.8$ |
| plate-slide-side-v2 | $2910.3 \pm 616.1$ | $2989.0 \pm 1044.1$ |
| plate-slide-back-v2 | $4378.5 \pm 373.0$ | $3963.4 \pm 927.3$ |
| plate-slide-back-side-v2 | $3872.0 \pm 1151.6$ | $4186.5 \pm 772.6$ |
| soccer-v2 | $443.5 \pm 785.5$ | $480.4 \pm 994.8$ |
| push-wall-v2 | $873.9 \pm 1718.3$ | $705.2 \pm 1535.7$ |
| shelf-place-v2 | $204.5 \pm 714.3$ | $366.1 \pm 919.0$ |
| sweep-into-v2 | $1297.9 \pm 1661.5$ | $506.2 \pm 1368.5$ |
| sweep-v2 | $1397.1 \pm 1922.3$ | $599.1 \pm 1359.2$ |
| window-open-v2 | $1453.6 \pm 1101.4$ | $3095.5 \pm 751.1$ |
| window-close-v2 | $2963.9 \pm 875.3$ | $3177.6 \pm 737.8$ |
| assembly-v2 | $2470.7 \pm 1758.6$ | $663.7 \pm 28.7$ |
| button-press-topdown-wall-v2 | $1270.9 \pm 214.5$ | $1199.9 \pm 203.7$ |
| hammer-v2 | $868.7 \pm 983.4$ | $1024.4 \pm 1024.3$ |
| peg-unplug-side-v2 | $1439.1 \pm 1817.8$ | $136.4 \pm 508.6$ |
| reach-wall-v2 | $4249.9 \pm 536.0$ | $4191.4 \pm 412.9$ |
| stick-push-v2 | $1288.5 \pm 1587.3$ | $790.1 \pm 1097.6$ |
| stick-pull-v2 | $601.4 \pm 1346.3$ | $287.1 \pm 922.8$ |
| box-close-v2 | $2683.7 \pm 1823.4$ | $2273.3 \pm 1707.3$ |

