# OpenReview forum: "Diffusion Model is an Effective Planner and Data Synthesizer for Multi-Task Reinforcement Learning"
_NeurIPS.cc/2023/Conference — NeurIPS 2023 poster_

### Official Review · Reviewer_gBBv · 2023-06-29

**Soundness:** 3 good
**Presentation:** 4 excellent
**Contribution:** 4 excellent
**Rating:** 7
**Confidence:** 4

**Summary:**

This paper proposed Multi-Task Diffusion Model, a diffusion-based method that incorporates Transformer backbones and prompt learning for generative planning and data synthesis in multitask offline settings. The performance of the proposed model on Meta-World and Maze2D benchmarks was shown.

**Strengths:**

This paper is the first to achieve both effective planning and data synthesis for multi-task RL via diffusion model and GPT.
This paper is well written and easy to follow.


**Weaknesses:**

# The difference between the proposed model and PromptDT is not well explained.
# Additional environment and ablation experiments may be more convincing.

**Questions:**

# PromptDT showed performance on MuJoCo. How is the performance of the proposed model on MuJoCo?
# This paper seemed to replace transformer in PromptDT with diffusion model. It’s better to explain the difference between the proposed model and PromptDT.

**Limitations:**

The related work was introduced in Section 4. It would be clearer to introduce related works before Section 2 Preliminaries.

---

> ### Author Rebuttal · Authors · 2023-08-10
>
> Thank you for your valuable comments and a positive assessment of our work! We are glad that you find our paper easy to follow, well-written, and the first to achieve both effective planning and data synthesis for multi-task RL. To address your concerns, we have added additional experiments on MuJoCo to validate the efficacy of our method. Our detailed response follows:
>
> >1. The difference between the proposed model and PromptDT is not well explained.
>
> Thank you for this good question! We would enrich the appendix with a dedicated section discussing their differences in the next version of our paper. The remarkable superiority of *MTDiff-p* over PromptDT emerges from our elegant incorporation of transformer architecture and trajectory prompt within the **diffusion model framework**, effectively modeling the multi-task trajectory distribution. PromptDT is built on Decision Transformer and it is trained in an autoregressive manner, which is limited to predicting actions step by step. However, *MTDiff-p* leverages the potency of sequence modeling, empowering it to adeptly perform trajectory generation. *MTDiff-p* has demonstrated SOTA performance in both multi-task decision-making and data synthesis empirical experiments, while PromptDT fails to contribute to data synthesis. Technically, *MTDiff-p* extends Decision Diffuser [1] into the multi-task scenario, utilizing classifier-free guidance for generative planning to yield high expected returns, which is also recognized by Reviewer EBNs.
>
> [1] Anurag Ajay, et al. Is conditional generative modeling all you need for decision making? International Conference on Learning Representations, 2023.
>
> >2. PromptDT showed performance on MuJoCo. How is the performance of the proposed model on MuJoCo?
>
> We have added new experiments to compare our method with PromptDT on the MuJoCo benchmark. We trained our model on the publicly available PromptDT datasets, i.e., *Cheetah-vel* and *Ant-dir*. These chosen environments have been judiciously selected due to their inherent diversity of tasks, serving as a robust test to validate the capability of multi-task learning. We report the scores (mean and std for 3 seeds) as follows:
>
> | Methods  | Cheetah-vel      | Ant-dir          |
> | -------- | ---------------- | ---------------- |
> | MTDiff-p | $-29.09\pm 0.31$ | $602.17\pm 1.68$ |
> | PromptDT | $-34.43\pm 2.33$ | $409.81\pm 9.69$ |
>
> We observed that *MTDiff-p* outperforms PromptDT largely, demonstrating its high efficacy and potency.

---

> > ### Comment · Reviewer_gBBv · 2023-08-18
> >
> > Thanks the authors for the detailed response and additional performance experiments. I’ll keep my score “accept".

---

> > > ### Author Response · Authors · 2023-08-18
> > > **Thank you!**
> > >
> > > We sincerely thank you for your recognition of our work! We really appreciate your effort to review our paper and your valuable comments! Thanks a lot.

---

> ### Comment · Area_Chair_qVd4 · 2023-08-17
>
> Dear Reviewer,
>
> The author has posted their rebuttal, but you have not yet posted your response. Please post your thoughts after reading the rebuttal and other reviews as soon as possible. All reviewers are requested to post this after-rebuttal-response.

---

### Official Review · Reviewer_nEaR · 2023-07-01

**Soundness:** 2 fair
**Presentation:** 3 good
**Contribution:** 3 good
**Rating:** 6
**Confidence:** 5

**Summary:**

The paper studies the use of diffusion models in offline multi-task reinforcement learning for planning and synthetic data generation. Both approaches use prompting to encode task-specific conditions for the generative model along with a transformer backbone. In the multitask setting, the approach outperforms prior SOTA baselines on MT50. In the synthetic data setting, the approach improves downstream TD3+BC performance.


**Strengths:**

- Clear and well-written presentation of the method
- A thorough set of baselines and strong evaluation on the challenging MT50 benchmark for MTDIFF-p
- Elegant method of removing the requirement of one-hot task encoding by transition prompting
- Strong results showing positive multi-task transfer in both planning and synthetic data generation, the algorithm neatly extends single-task equivalents in both areas


**Weaknesses:**

- Only custom settings with large amounts of data are considered, it would be useful to understand what the minimum data needed to be effective is. Additionally, it would be useful to evaluate on pre-existing offline datasets for a more representative comparison.
- In Figure 6, it would be useful to compare MTDIFF-s-single with the existing single task baseline, Synthetic Experience Replay. [1]
- More thorough analysis of data quality would be valuable rather than just downstream performance, e.g. error of the transitions. Furthermore, downstream RL performance is only computed with one RL algorithm.
- The two parts of the paper - multitask planning and synthetic data generation - are not necessarily connected. For example, the synthetic data (MTDIFF-s) is used for an entirely different RL algorithm, TD3+BC, and does not contribute to the performance of the planning algorithm (MTDIFF-p).

Minor:
- Line 242: Typo ‘fine-grind’
- Line 248: the author’s definition of ‘near-optimal’ is the same as D4RL ‘full-replay’ and ‘sub-optimal’ is the same as D4RL ‘medium-replay’. These descriptions may be more clear to offline RL practitioners
- Figure 4: choosing one sampled trajectory for each approach is likely cherry-picking, I would suggest rendering N samples
- Figure 5: missing standard deviation
- Figure 6: missing the baseline without synthetic data
- Line 15: Unclear what ‘high-quality’ and ‘low-quality’ mean as it seems MTDIFF-s models the original distribution

[1] Synthetic Experience Replay. Cong Lu, Philip J. Ball, Yee Whye Teh, Jack Parker-Holder.


**Questions:**

- How would the prompting perform if the tasks were not readily discernible from the initial transitions? E.g. multi-task settings with sparse task-specific reward
- What is the minimum data required to make the algorithm work?
- What is the speed of planning and sampling transitions with the transformer model? How does this compare to related methods?
- How does MTDIFF-s-single compare to Synthetic Experience Replay? I.e. what is the benefit of modeling full trajectories instead of transitions?


**Limitations:**

Discussion of limitations hidden in Appendix, should be moved to the main paper.

---

> ### Author Rebuttal · Authors · 2023-08-10
>
> >1. ... custom settings with large amounts of data ... minimum data needed to be effective is.
>
> We downsample the "near-optimal dataset" to 0.1$\times$/ 0.2$\times$/ 0.3$\times$ the size via random selection. We observed that the performance of *MTDiff-p* decreases with dataset reduced, dropping at $36.06\%\pm2.36\%$ success rate for 0.1$\times$ size data, which can be caused by missing parts of optimal trajectories. To validate that, we re-train *MTDiff-p* on 0.1$\times$ size data containing *only* expert trajectories. In this case, *MTDiff-p* can even obtain $85.05\%\pm1.54\%$ success rate. It is reasonable that the most critical factor for goal-conditional planning is the optimality of the dataset instead of data quantity, which has also been verified in [1]. In our paper, we consider the more challenging setting, i.e., using data with low return accounting for majority of the dataset.
>
> >2. evaluate on pre-existing offline datasets for a more representative comparison.
>
> Our method aims to learn a diffusion model from multi-task offline data. However, pre-existing datasets like D4RL only contain single-task datasets in each domain. As a result, we collect multi-task datasets for evaluation. Meta-world is chosen since it contains 50 various and difficult tasks, and it is well-known for its efficiency. For reproductivity, we give the code in supplementary materials and the link to our dataset in Appendix. We remark that we added new experiments on Mujoco and *MTDiff-p* still achieves the SOTA performance, where the details can be seen in our response to Reviewer *gBBv*.
>
> >3. In Figure 6, it would be useful to compare MTDIFF-s-single with the existing single task baseline.
>
> We remark that SynthER [2] and MTDiff are almost concurrent work. Fig. 6 is used to support our hypothesis that *MTDiff-s* benefits from multi-task training instead of how *MTDiff-s-single* outperforms other methods. To compare *MTDiff-s-single* with SynthER, we borrowed the open-sourced code and trained SynthER on coffee-push, disassemble, and dial-turn tasks. We added the result to Fig. 6, and the revised vision is referred to Fig. 2 in **Global PDF**. We find that SynthER still underperforms *MTDiff-s*.
>
> >4. More thorough analysis of data quality ... downstream RL performance is only computed with one RL algorithm.
>
> We refer to **Global PDF** for the new experiments and explanations.
>
> >5. ...multitask planning and synthetic data generation are not necessarily connected...
>
> Thanks for your question. Indeed, MTDIFF-s can augment the multi-task dataset to benefit MTDIFF-p, while it can be biased in comparison. First, MTDIFF-p is a generative planning method and its performance is closely connected to the data optimality rather than the data coverage (see [1] and R1). MTDIFF-s is designed to improve the data coverage and it is more suitable to use offline RL for evaluation. Second, for a specific task, even MTDIFF-p's performance improves with data augmentation by MTDIFF-s, it can be caused by the augmented data of other tasks since MTDIFF-p can conduct implicit knowledge sharing among tasks. As a result, we conduct single-task augmentation and use offline RL for evaluation.
>
>
> >6. Minor error
> * We will fix the typo in the next version.
> * We remark that definition of "near-optimal" and "sub-optimal" comes from paper [3], and we would mention replay and medium-replay based on your suggestion.
> * We show more rendered cases for Maze2D in Appendix D, however, we added another example (see Fig. 4 in **Global PDF**) to demonstrate the effectiveness of our method.
> * The corrective Fig. 5 is referred to Fig. 1 in **Global PDF**.
> * We added the results of the baseline without synthetic data into Fig. 6, which is referred to Fig. 2 in **Global PDF**.
> * A good quality means the generated data has (1) low dynamics errors to follow the true dynamics of the original task, and (2) extended data coverage to enlarge the dataset. Fig. 4 in appendix shows the synthetic data expands the coverage of the original data, thus boosting the performance of offline RL.
>
> >6. How would the prompting perform if the tasks were not readily discernible from the initial transitions?
>
> Thanks for the question! It is possible that the tasks cannot be identified from their initial transitions. Nevertheless, the prompt contains both the initial transitions and the task-specific label $Z$, where $Z=(s^*_i,a^*_i,\ldots,s^*\_{i+J-1},a^*\_{i+J-1})$ is the trajectory prompt sampled from an expert trajectory to provide task-identified information. $Z$ is injected into the model as a condition during both training and testing.
>
> >7. ...speed of planning and sampling ...? How does this compare to related methods?
>
> It is known that the sampling speed of diffusion model according to the diffusion process can be quite slow, which we have discussed in Appendix F. For a concrete example in MetaWorld, it takes on average 1.9s in wall-clock time to generate one action sequence for planning (hardware being a 3090 GPU). We can improve the inference speed by leveraging a recent sampler called DPM-solver [4] to decrease the diffusion steps required to 0.2$\times$ without any loss in performance, and using a larger batch size (leveraging the parallel computing power of GPUs) to evaluate multiple environments at once. Thus the evaluation run-time roughly matches the runtime of non-diffusion algorithms (diffusion step is 1). We will add an section to discuss the sampling speed of our method in Appendix in the next version.
>
> >8. Discussion of limitations ... moved to main paper.
>
> We would move it into the Conclusion section in the next version.
>
> [1] Rethinking Goal-Conditioned Supervised Learning and Its Connection to Offline RL. ICLR 2022
>
> [2] Synthetic Experience Replay. ICLRW 2023
>
> [3] Offline Q-learning on Diverse Multi-Task Data Both Scales And Generalizes. ICLR 2023
>
> [4] DPM-Solver: A Fast ODE Solver for Diffusion Probabilistic Model Sampling in Around 10 Steps. NeurIPS 2022

---

> > ### Comment · Reviewer_nEaR · 2023-08-13
> > **Thank you**
> >
> > Thank you to the authors for the clarifying response and additional experiments. I have raised my score accordingly.

---

> > > ### Author Response · Authors · 2023-08-14
> > > **Thank You!**
> > >
> > > We sincerely thank you for your recognition and for increasing your score! We really appreciate it!
> > >
> > > The constructive suggestions you gave during the rebuttal session are greatly helpful in improving the quality of our paper. Thanks to your time and hard work!

---

### Official Review · Reviewer_UScg · 2023-07-06

**Soundness:** 3 good
**Presentation:** 3 good
**Contribution:** 3 good
**Rating:** 6
**Confidence:** 5

**Summary:**

- The paper investigates the effectiveness of learning a diffusion model for modelling multi-task offline data. To do so, the paper introduces two variants of using a learned diffusion model: (a) by planning over a sequence of actions, (b) generating data, and using the generated data for offline policy optimisation/improvement.

- The paper compares the proposed method to various different baselines, and shows decent improvements over all the baselines.

- The paper shows the ability of the proposed model to generate useful data, by augmenting kow-quality datasets.

- The paper can be seen as an extension of Decision Diffuser for multi-task scenario.



**Strengths:**

- The paper is very well written.
- The paper does a good job comparing to various strong baselines, as well as showing the usefulness of the generated synthetic data.
- The proposed method is shown to be effective planner for solving multi-task problem on Maze2D and Meta-World benchmarks.

**Weaknesses:**

No such weakness as such.

**Questions:**

- For Table H in appendix, it will be useful to report results for other baselines too.
- It will be useful to see how the performance changes as a result of varying the generated data. For example: MTDIFF-S synthesizes 2M transitions to expand the original 1M dataset. It will be helpful to report reports as to how the baselines perform by varying amount of data generated or augmented.
- It will also be helpful to see how the performance changes as the MTDIFF-S  is trained on less amount of tasks (i.e., instead of 45 tasks it's trained on 5/10/15/20/30 tasks, as the working hypothesis is MTDIFF-S can perform implicit data sharing, so will be useful to verify this by decreasing the number of tasks (figure 6 only shows the comparison for 2 tasks).

**Limitations:**

See Weaknesses and Questions.

---

> ### Author Rebuttal · Authors · 2023-08-10
>
> Thank you for your detailed feedback and for a positive assessment of our work! We carefully address your concerns as follow:
>
> >1. For Table H in appendix, it will be useful to report results for other baselines too.".
>
> Thanks for this good suggestion! Considering the space limitation during this rebuttal period, we would report the results in the next version of our paper.
>
> >2. It will be useful to see how the performance changes as a result of varying the generated data.
>
> Thank you for this constructive suggestion! We re-run the experiments on coffee-push and disassemble tasks, expanding the original 1M dataset with 0.5M, 1M and 2M generated data for each task respectively. Then we continue to use TD3-BC as the downstream algorithm to measure the policy improvement. The result shows the performance increases along with more augmented data. Meanwhile, we find our method with only 0.5M generated data even outperforms RAD and S4RL which use 2M and 10M augmented data, respectively.
>
>
> | Tasks       | 2M Generated    |  1M Generated  | 0.5M Generated | Origin          |
> | ----------- | --------------- | :------------: | :------------: | --------------- |
> | coffee-push | $74.67\pm 6.79$ | $70.66\pm7.78$ | $65.43\pm9.91$ | $28.60\pm14.55$ |
> | disassemble | $69.00\pm 4.72$ | $63.6\pm4.62$  | $60.83\pm4.40$ | $60.20\pm16.29$ |
>
> >3. It will also be helpful to see how the performance changes as the MTDIFF-s is trained on less amount of tasks.
>
> Thank you for this constructive suggestion! We re-train *MTDiff-s* on 30/20/10 tasks respectively,  and then measure the policy improvement on the coffee-push task across 3 seeds to verify our hypothesis that multi-task training enables implicit data sharing in *MTDiff-s*.
>
> Our findings, as outlined below, provide compelling evidence in support of our hypothesis: *MTDiff-s* exhibits progressively superior data synthesis performance with increasing task diversity.
>
> | Number of Taks | coffee-push     |
> | -------------- | --------------- |
> | 45             | $74.67\pm 6.79$ |
> | 30             | $68.33\pm9.00$  |
> | 20             | $66.53\pm7.12$  |
> | 10             | $65.06\pm 9.39$ |
> | 1              | $63.41\pm8.99$  |

---

> > ### Comment · Reviewer_UScg · 2023-08-17
> > **Thank you**
> >
> > Thank you for running extra experiments.

---

> > > ### Author Response · Authors · 2023-08-17
> > > **Thank you！**
> > >
> > > Thank you for your acknowledgment of the extra experiments we conducted! We really appreciate your effort to review our paper and your recognition of our work. We are glad to see that our paper's quality has greatly improved with your valuable suggestions. If you have any additional suggestions or questions, please feel free to let us know.

---

### Official Review · Reviewer_EBNs · 2023-07-07

**Soundness:** 3 good
**Presentation:** 3 good
**Contribution:** 3 good
**Rating:** 6
**Confidence:** 3

**Summary:**

This paper extends diffusion-based planners to multi-task settings by combining prompt learning. Specifically, a few segments from expert demonstrations are used as task prompts to distinguish different tasks and guide the diffusion model to generate task-specific trajectories. A classifier-free guidance approach is used to sample trajectories that yield high expected returns. Unlike previous diffusion-based planners, the authors utilize a Transformer as the denoising network. The experiments conducted demonstrate a marked improvement in performance.

**Strengths:**

This paper is easy to follow. I feel quite enjoyable while reading this paper.

**Weaknesses:**

- The experiment section needs further revision:
  - Figure 5 x-label is not correct.
- What are the quantitative results on Maze2D unseen map
- Figure 6, what is the criterion for selecting these 2 tasks? This experiment compares the model training from scratch and the model with pre-training. The results are not that surprising since pre-training provides a better initialization in general.
- line 334: The subtitle is confusing. This paragraph is mainly about figure 6, not comparing other augmentation methods.
- In Table 2, can the authors add results with 2M random data expanded for S4RL and RAD? Just want to remove the effect of different data sizes.
- In Table 2, what are the original results? Is it the maximal return of the dataset?

**Questions:**

Please see weaknesses.

**Limitations:**

Yes.

---

> ### Author Rebuttal · Authors · 2023-08-10
>
> We thank the reviewer for their thorough and detailed review and welcome suggestions for improvement.  Here, we address your concerns as follows:
>
> >1. The experiment section needs further revision: Figure 5 x-label is not correct.
>
> We apologize for mistaking the x-label of Fig. 5. This figure quantifies the average scores obtained in 8 training maps of MTDiff-p and PromptDT, respectively. The final corrected version of Figure 5 is referred to Fig. 1 in the global PDF (with standard deviation added).
>
> >2. What are the quantitative results on Maze2D unseen map？
>
> Thank you for this question! We have added another unseen map for thorough comparison, which is rendered in Fig. 4 within the global PDF. Subsequently, we evaluate *MTDiff-p* and PromptDT on the 2 unseen maps for 100 episodes (150 steps for an episode), and report average scores as follows:
>
> |   Method   | Scores  |
> | :--------: | :-----: |
> | *MTDiff-p* | $23.37$ |
> |  PromptDT  | $17.03$ |
>
> >3. Figure 6, what is the criterion for selecting these 2 tasks? This experiment compares the model training from scratch and the model with pre-training. The results are not that surprising since pre-training provides a better initialization in general.
>
> We would like to provide further clarification regarding Figure 6. It is essential to note that the figure does not illustrate the model acquired through pre-training; rather, it serves as a comparative representation of policies learned with distinct synthetic datasets generated by single-task and multi-task models.
>
> Concretely, we train *MTDiff-s* on the multi-task dataset encompassing 45 tasks and *MTDiff-s-single* on the single-task dataset. The models, *MTDiff-s* and *MTDiff-s-single*, are then leveraged to generate task-specific data (i.e., for *coffee-push* and *disassemble* in Fig. 6) for data augmentation in offline RL learning. We compare the performance trained on these two kinds of augmented data, and find *MTDiff-s* shows a significant advantage compared to *MTDiff-s-single*. The reason would be that *MTDiff-s* can conduct implicit knowledge sharing [1] among tasks and transfer knowledge from other tasks to expand the data coverage of the target task, which is also appreciated by Reviewer 8zHR and UScg. As a result, both methods in Fig. 6 are trained from scratch with augmented data without pre-training.
>
> The reason why we select these 2 tasks is that they are difficult and the policies perform relatively poorly learned with their original dataset. They are suitable for evaluating policy improvement after data augmentation to validate our hypothesis. We also added new experiments on another new task *dia-turn* and observed similar results, which is referred to Fig. 2 in the global PDF.
>
> >3. The subtitle is confusing.
>
> This paragraph highlights the superior performance of *MTDiff-s* in comparison to the baseline methods and offers a comprehensive analysis of the advantageous aspects derived from its multi-task training paradigm. To enhance clarity, we propose revising the subtitle to read:  "How does *MTDiff-s* perform and benefit from multi-task training?"
>
> >4. In Table 2, can the authors add results with 2M random data expanded for S4RL and RAD?
>
> Thanks for the question. We would like to clarify that both S4RL and RAD have already used additional augmented data in their training process, so it is no need to add more samples to the dataset. Specifically, during each training step of S4RL or RAD, a batch of transitions are sampled and the algorithm augments the transitions to generate new ones. As a result, the size of newly added training samples is batch_size $\times$ training_steps (e.g., 400$\times$1M), which is much larger than the 2M augmented data in *MTDiff-s*.
>
> >5. In Table 2, what are the original results? Is it the maximal return of the dataset?.
>
> The original result is the success rate attained by the TD3-BC agent following training on the original 1 million (1M) dataset.
>
> [1] Tianhe Yu, et al. Conservative data sharing for multi-task offline reinforcement learning. NeurIPS 2021.

---

> > ### Comment · Reviewer_EBNs · 2023-08-17
> > **Response**
> >
> > Thank the authors for the clarification, especially in Figure 6. I'm happy to increase my score from 4 to 6.

---

> > > ### Author Response · Authors · 2023-08-18
> > > **Thank you!**
> > >
> > > We sincerely thank you for engaging with us and for increasing your score! We really appreciate your effort to review our paper and your recognition of our work!
> > >
> > > The constructive suggestions you gave during the rebuttal session are greatly helpful in improving the quality of our paper. Thanks again for your time and hard work!

---

### Official Review · Reviewer_8zHR · 2023-07-08

**Soundness:** 3 good
**Presentation:** 2 fair
**Contribution:** 3 good
**Rating:** 7
**Confidence:** 4

**Summary:**

The author propose a diffusion model MTDIFF for multi-task RL. The goal is to leverage diffusion process and transformer backbone to have a sota generative model for RL. The author demonstrates its effectiveness with generative planning on meta world and data augmentation.

**Strengths:**

Combing transformer and diffusion together is a straightforward idea, but the paper excuted it very well. Also leveraging the power of prompting in generative planning is a cool idea. The analysis on how MTDIFF benefit from multi-tasking in data synthesis is interesting.

**Weaknesses:**

The paper also seems unpolished. On table 1, row MTIQL is empty. Also the figure 5 should be "We evaluate these two methods on both seen maps and an unseen map. The average scores obtained in 8 training maps are referred to Figure 5", but I don't know which is PromptDT which is yours.

**Questions:**

With generative planning, can the authors show the visualization of how model get from random noise to optimal trajectory? Like a GIF would be nice.

---

> ### Author Rebuttal · Authors · 2023-08-10
>
> We thank the reviewer for their valuable comments and a positive assessment of our work! We are glad that they found that our paper provides an interesting analysis and executes a straightforward idea very well. To address your concerns, we would polish our paper in the next version to make it clear, and our detailed response follows:
>
> >1. On table 1, row MTIQL is empty.
>
> In Table 1, the row of **MTCQL** remains empty as MTCQL almost fails on the Metaworld MT50-v2 benchmark. The corresponding analysis is in lines 307-309 in our paper. We hypothesize that the failure of MTCQL emanates from penalizing the O.O.D. actions across diverse tasks explicitly, thereby exacerbating the challenge of distribution shift within the ambit of multi-task training. Through our empirical experiments, we observed that MTCQL would firstly achieve around 10% success rate at the early training stage, and then drop down to almost 0% success rate.
>
> >2. Also the figure 5 should be "We evaluate these two methods on both seen maps and an unseen map. The average scores obtained in 8 training maps are referred to Figure 5", but I don't know which is PromptDT which is yours.".
>
> We are sorry that we mistook the x-label of Fig. 5. This figure quantifies the average scores obtained in 8 training maps of MTDiff-p and PromptDT, respectively. The final corrected version of Figure 5 is referred to Fig. 1 in the global PDF (with standard deviation added).
>
> >3. With generative planning, can the authors show the visualization of how model get from random noise to optimal trajectory? Like a GIF would be nice.
>
> Thank you for this good suggestion! We present a visual depiction of the model's performance improvement as the number of denoised steps increases on Maze2D, and the result can be seen in Fig. 3 in the global PDF.
>
> We appreciate the reviewer’s feedback again which helped us to improve the quality of the paper. We also hope that our response have sufficiently addressed the reviewer’s concerns.

---

> > ### Comment · Reviewer_8zHR · 2023-08-18
> > **Thanks for rebuttal**
> >
> > I ve read the rebuttal and my score maintains

---

> > > ### Author Response · Authors · 2023-08-18
> > > **Thank you**
> > >
> > > We sincerely thank you for your recognition of our work! We really appreciate your effort to review our paper and your valuable comments! Thanks a lot.

---

> ### Comment · Area_Chair_qVd4 · 2023-08-17
>
> Dear Reviewer,
>
> The author has posted their rebuttal, but you have not yet posted your response. Please post your thoughts after reading the rebuttal and other reviews as soon as possible. All reviewers are requested to post this after-rebuttal-response.

---

### Author Rebuttal · Authors · 2023-08-10

### General Response

We thank all of the reviewers for their time and insightful comments. Furthermore, we are very glad to find that reviewers generally recognized our key contributions and clear presentation of our paper:

#### Contributions:

* **Method:** "This paper is the first to achieve both effective planning and data synthesis for multi-task RL via diffusion model and GPT" [gBBv]. The proposed method is shown to be effective and positive, which extends previous single-task models to solving more complex multi-task problems [UScg, nEaR]. "Leveraging the power of prompting instead of one-hot task encoding is a cool and elegant idea" [8zHR, nEaR]. "The analysis of how MTDIFF benefits from multi-tasking in data synthesis is interesting" [8zHR].
* **Experiment:** The paper does a good job comparing various strong baselines, and shows decent improvements over all the baselines [UScg]. The experiments conducted demonstrate a marked improvement in performance [EBNs]. The paper shows a thorough set of baselines and the approach outperforms prior SOTA baselines [nEaR].
* **Presentation:** The paper is well written [UScg, gBBv]. The paper shows clear and well-written presentation of the method [nEaR]. The paper is easy to follow [EBNs, gBBv].

Meanwhile, we thank all the reviewers for their helpful and constructive feedback to improve the quality of our work. In addition to the pointwise responses below, we would carefully update our paper in the next version to incorporate the valuable suggestions of the reviewers:

* [**UScg**] We would add the experimental results about varying the generated data of *MTDiff-s* and varying the trained tasks of *MTDiff-s* (see the response to Reviewer UScg).
* [**UScg**] We would report performance for baselines on each single task in MT50-v2.
* [**nEaR**] We would add one more unseen Maze2D map where *MTDiff-p* succeeds while PromptDT fails (see Fig. 4 in global PDF).
* [**nEaR**] We would update Figure 6 to compare with more baselines (see Fig. 2 in global PDF).
* [**nEaR**] We would polish some of our expressions (e.g. definition) and add a section to discuss the sampling speed of our model (see the response to Reviewer nEaR).
* [**nEaR**] We would add experimental results about data quality analysis and IQL performance improvement (see Table 1 and Table 2 in the global PDF).
* [**EBNs**] We would revise the subtitle the line 334 (see the response to Reviewer EBNs).
* [**EBNs**] We would add quantitative results on Maze2D unseen map (see the response to Reviewer EBNs).
* [**8zHR, EBNs, nEaR**] We would update Figure 5 (see Fig. 1 in global PDF).
* [**gBBv**] We would add a section in the appendix to demonstrate the difference between our model and PromptDT, and add the experimental results on MuJoCo benchmark (see the response to Reviewer gBBv).

We hope to have addressed all the raised concerns and would be happy to respond to further questions and suggestions.

---

### Decision · Program_Chairs · 2023-09-21

**Decision:**

Accept (poster)

**Comment:**

The paper proposes Multi-Task Diffusion Model (MTDiff), a method that integrates Transformers and diffusion models for generative planning and data synthesis in multi-task offline reinforcement learning settings. It demonstrates the efficacy of MTDiff in a variety of benchmarks, including Meta-World and Maze2D, for both planning and synthetic data generation tasks.

Reviewers generally commend the well-executed integration of Transformers and diffusion models, along with the innovative use of prompting in generative planning. The paper also stands out for its clear writing and strong evaluation against various baselines. However, there are areas for improvement such as the unpolished initial version, as pointed out in the figure and table issues. A more thorough analysis on data quality would also add value, and there is a disconnect between the paper's two main components - multi-task planning and synthetic data generation. Some of these weaknesses have been addressed during the rebuttal stage. Given these considerations, the paper is recommended for acceptance with the understanding that the authors will make the necessary revisions.